# Compensatory metabolic networks in pancreatic cancers upon perturbation of glutamine metabolism

Douglas E. Biancur[1,2,*], Joao A. Paulo[3], Beata Małachowska[4,5], Maria Quiles Del Rey[1], Cristovão M. Sousa[1], Xiaoxu Wang[1], Albert S.W. Sohn[2], Gerald C. Chu[6], Steven P. Gygi[3], J. Wade Harper[3], Wojciech Fendler[1,4], Joseph D. Mancias[1,3,*] & Alec C. Kimmelman[1,2]

Pancreatic ductal adenocarcinoma is a notoriously difficult-to-treat cancer and patients are in need of novel therapies. We have shown previously that these tumours have altered metabolic requirements, making them highly reliant on a number of adaptations including a non-canonical glutamine (Gln) metabolic pathway and that inhibition of downstream components of Gln metabolism leads to a decrease in tumour growth. Here we test whether recently developed inhibitors of glutaminase (GLS), which mediates an early step in Gln metabolism, represent a viable therapeutic strategy. We show that despite marked early effects on *in vitro* proliferation caused by GLS inhibition, pancreatic cancer cells have adaptive metabolic networks that sustain proliferation *in vitro* and *in vivo*. We use an integrated metabolomic and proteomic platform to understand this adaptive response and thereby design rational combinatorial approaches. We demonstrate that pancreatic cancer metabolism is adaptive and that targeting Gln metabolism in combination with these adaptive responses may yield clinical benefits for patients.

[1] Division of Genomic Stability and DNA Repair, Department of Radiation Oncology, Dana-Farber Cancer Institute, Boston, Massachusetts 02215, USA. [2] Perlmutter Cancer Center, Department of Radiation Oncology, NYU Medical School, New York, New York 10016, USA. [3] Department of Cell Biology, Harvard Medical School, Boston, Massachusetts 02115, USA. [4] Department of Biostatistics and Translational Medicine, Medical University of Lodz, Lodz 91-738, Poland. [5] Postgraduate School of Molecular Medicine, Medical University of Warsaw, Warsaw 02-091, Poland. [6] Department of Pathology, Brigham and Women's Hospital, Harvard Medical School, Boston, Massachusetts 02115, USA. * These authors contributed equally to this work. Correspondence and requests for materials should be addressed to J.D.M. (email: Joseph_Mancias@dfci.harvard.edu) or to A.C.K. (email: Alec.Kimmelman@nyumc.org).

Pancreatic ductal adenocarcinoma (PDAC) is an extremely aggressive disease with poor prognosis[1]. Treatment options remain largely ineffective with a 5-year survival rate of <8% (ref. 2). Therefore, novel treatment options are essential. The driving oncogene in PDAC results from Kras mutation, which is present in >90% of human tumours[3–5]. Our group and others have shown that oncogenic Kras can promote a metabolic rewiring of PDAC, including the non-canonical use of glutamine (Gln) to support proliferation through redox homoeostasis[6–10]. The ability to cope with oxidative stress is a defining characteristic of many cancers[11–13] and the nutrient poor and hypoxic microenvironment of PDAC suggests that redox maintenance is essential for tumour proliferation. As Gln metabolism is dispensable for non-malignant cells, yet has a critical role in PDAC redox homoeostasis, it would seem to be an ideal candidate for therapeutic targeting[9].

The first step in glutaminolysis is mediated by the enzyme glutaminase (GLS), which catalyzes the conversion of Gln to glutamate (Glu) in the mitochondria where, in PDAC, Gln-derived Glu is metabolized ultimately resulting in increased reducing potential in the form of increased NADPH and glutathione (GSH)[9]. The disruption of Gln metabolism, through inhibition of GLS (GLSi), results in decreased antioxidant response and decreased cell proliferation. Until recently, drugs targeting GLS suffered from poor bioavailability and low potency[14]. CB-839 is a potent orally bioavailable inhibitor of GLS but not liver-expressed GLS2 (ref. 15) that has shown efficacy in mouse models of cancer including Myc-driven hepatocellular carcinoma, lymphoma, and breast cancer but limited efficacy in lung cancer mouse models[16–18]. Ongoing early phase clinical trials are examining the efficacy of CB-839 in haematologic and solid cancers.

An outstanding question in PDAC is whether GLSi is a viable therapeutic strategy, given it is the most proximal enzyme in the PDAC-specific Gln metabolism pathway, and how this may differ from targeting distal parts of the pathway[9]. This may be particularly relevant as pancreatic tumours have the ability to scavenge for fuel sources through multiple processes including autophagy, macropinocytosis and uptake of free amino acids released by stromal cells[19–23]. In this study, through the integration of proteomics and metabolomic profiling, we characterize the adaptive mechanisms PDAC may utilize to mitigate effects of targeting metabolic pathways and provide a critical proof of concept that targeting these compensatory pathways may have therapeutic utility.

## Results

**GLSi has antiproliferative activity in PDAC cells.** Based on our earlier results, we first reassessed the role of Gln metabolism in PDAC cells by impairing GLS activity using RNA interference. GLS knockdown significantly reduced PDAC growth as shown previously by our group[9] (Fig. 1a, $P < 0.001$, $t$-test, Supplementary Fig. 1a). As prior results also showed an anti-proliferative effect of the GLS inhibitor BPTES (bis-2-(5-phenylacetamido-1,2, 4-thiadiazol-2-yl)ethyl sulfide) (Supplementary Fig. 1b), we were interested in examining the efficacy of CB-839, a higher potency and orally bioavailable GLS inhibitor[15]. CB-839 had a potent effect on the proliferation of multiple PDAC cell lines, including cells derived from primary tumours of a LSL-Kras$^{G12D}$; p53$^{L/+}$, Pdx1-Cre mouse model of PDAC (MPADC-4)[24], with IC50s in the low nanomolar range (Fig. 1b, c). In contrast, a mouse embryonic fibroblast cell line and a human lung fibroblast cell line (IMR90) were relatively insensitive to CB-839 (Supplementary Fig. 1c,d). CB-839 treatment significantly reduced PDAC growth across multiple cell lines, as tested in a multi-day proliferation assay (Fig. 1d,e and Supplementary Fig. 1e–h,

$P < 0.002$, $t$-test). To demonstrate the on-target activity of CB-839, we rescued proliferation of PDAC cell lines by adding excess Glu, the end product of GLS (Fig. 1d,e). PDAC lines were also sensitive to CB-839 treatment in RPMI media as well as in 3D-culture (Supplementary Fig. 1i,j). We noted that in contrast to prior reports of CB-839 in triple-negative breast cancer cell lines, PDAC cell lines did not exhibit cell death or apoptosis (Fig. 1f,g)[15]. We next examined the effect of GLSi on the levels of intracellular metabolites in PDAC cell lines. As expected, at 6 h post CB-839 treatment of MPDAC-4 and a human PDAC cell line, Gln accumulated and Glu levels were significantly decreased (Fig. 1h, Supplementary Fig. 1k, $P < 0.05$, for all, $t$-test). In addition, CB-839 reduced the levels of a number of metabolites downstream of Glu including aspartate (Asp), α-ketoglutarate (α-KG), GSH, malate and succinate.

**GLSi has no antitumour effect in a mouse model of PDAC.** Based on the anti-proliferative effect of CB-839 in PDAC cell culture systems, we were interested in testing the efficacy of GLSi *in vivo* in a treatment-resistant autochthonous mouse model of PDAC (LSL-Kras$^{G12D}$; p53$^{L/+}$; Pdx1-Cre) that closely mimics the human condition[20,24,25]. We first determined the pharmacokinetic and pharmacodynamics profile of CB-839 in mice with tumours identified via ultrasound[26]. CB-839 was administered at 200 mg kg$^{-1}$, a dose determined previously[15]. Tumour and plasma were collected 4 h after dosing and CB-839 concentrations of >2 nmol g$^{-1}$ or μmol l$^{-1}$ were observed (Fig. 2a). This was associated with a significant suppression of GLS activity in tumours (Fig. 2b, $P = 0.01130$, $t$-test) and the expected changes in metabolites (Fig. 2c). We subsequently performed a mouse clinical trial testing CB-839 versus vehicle control with overall survival as the primary end point. We also monitored tumour growth via serial ultrasound as a secondary end point. As shown in Fig. 2d, there was no significant improvement in survival with CB-839 with mice treated with CB-839 actually showing marginally shorter median survival time than vehicle-treated animals (20 versus 24 days, $P = 0.3441$). Furthermore, mice treated with CB-839 developed secondary pancreatic tumours ($P = 0.0335$) and had a trend towards increased macrometastatic disease ($P = 0.1819$) (Supplementary Fig. 2a,b). We noted secondary tumour development as early as 5 days post initiation of CB-839 (Supplementary Fig. 2c). Owing to the rapid kinetics of secondary tumour formation with GLSi, we reasoned that secondary tumours arose from pre-existing either late-stage pancreatic intraepithelial lesions or early PDAC (as opposed to in-transit metastasis) not detectable via ultrasound but likely present owing to the nature of the mice with pancreas-wide expression of oncogenic Kras and p53 heterozygosity. Given secondary tumour development in a majority of CB-839-treated mice, the growth kinetics of the primary tumour could not be accurately determined via ultrasound. Primary tumours did appear to progress despite CB-839 treatment based on initial ultrasound measurement and direct tumour measurements at autopsy. Secondary tumours and macrometastatic disease were confirmed as adenocarcinoma, did not appear morphologically distinct from the primary, and had adequate drug levels and GLSi (Supplementary Fig. 2d–j). We tested for markers of epithelial-mesenchymal transition (EMT) to determine whether CB-839 treatment initiated an EMT phenotype that could account for the increase in metastases. EMT markers from primary tumours were highly variable likely reflective of cellular heterogeneity within the tumours (Supplementary Fig. 2k). Although EMT markers were slightly elevated with CB-839 treatment in cell culture,

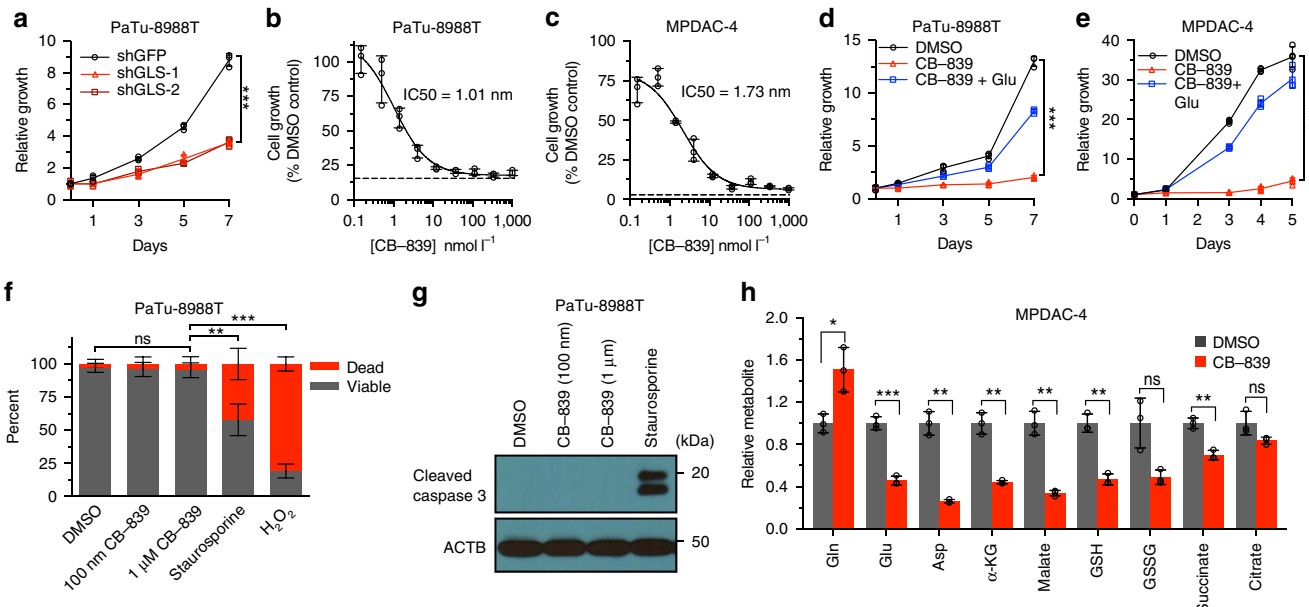

**Figure 1 | GLSi with CB-839 has antiproliferative activity in PDAC cell lines. (a)** Relative proliferation of PaTu-8988T cells expressing a control shRNA or shRNAs to GLS (#1 and #2). Data are plotted as relative cell proliferation in arbitrary units. Error bars depict ± s.d. of three independent wells from a representative experiment (of four experiments). **(b,c)** Cell proliferation dose–response curves for PaTu-8988T (**b**) and MPDAC-4 (**c**) treated with CB-839 for 72 h. The dashed line indicates the relative reporter signal at the time of CB-839 addition. Error bars depict ± s.d. of three independent wells from a representative experiment (of three experiments). **(d,e)** Relative proliferation of PDAC cell lines, PaTu-8998T (**d**) and MPDAC-4 (**e**) treated with CB-839 (100 nm) or DMSO. As indicated, glutamate (Glu, 4 mM) was added to the media at the time of CB-839 addition. Error bars depict ± s.d. of three (**d**) or four (**e**) independent wells from a representative experiment (of four experiments). **(f)** PaTu-8988T cell viability as determined by trypan blue exclusion assay at 72 h after DMSO and CB-839. Staurosporine (1 μm) and $H_2O_2$ (1 mM) positive controls treated for 3 h. Data are represented as mean ± s.d. of three independent wells from a representative experiment (of four experiments). **(g)** Cleaved Caspase-3 western blotting analysis of extracts from PaTu-8988T cells at 72 h after DMSO or CB-839. Staurosporine positive controls treated for 3 h. **(h)** Relative metabolite abundance in MPDAC-4 cells following 6 h CB-839 treatment. Data are presented as mean total metabolite pools ± s.d. of three independent wells from a representative experiment (of three experiments). For all panels, significance determined with $t$-test. $*P < 0.05$, $**P < 0.01$, $***P < 0.001$, ns: non-significant, $P > 0.05$.

together these results do not consistently support a clear role for EMT in this process (Supplementary Fig. 2l). Although efforts (Supplementary Fig. 2, Methods) at determining the cause of this phenomenon were unrevealing, the nature of the model with ubiquitous pancreatic expression of oncogenic Kras will allow for future studies which may reveal the molecular basis of this phenomenon. Given the paradoxical effect of GLSi in the autochthonous model, showing no objective tumour responses, despite the robust responses *in vitro*, we hypothesized that either the pancreatic tumour microenvironment or more simply the *in vivo* milieu may influence the metabolic response of PDAC.

To test these possibilities, we utilized transplantation models that would allow us to assess the impact of *in vivo* growth in different environments. We first examined the efficacy of CB-839 in an orthotopic model of PDAC. We implanted the highly CB-839 sensitive MPDAC-4 cell line (Fig. 1c,e) into the pancreata of nude mice and treated with CB-839. There was no significant tumour growth delay as monitored by luciferase imaging or end point tumour weight (Fig. 3a,b, $P = 0.3412$, $P = 0.8845$, respectively, $t$-test). To understand whether the marked shift in sensitivity to CB-839 treatment *in vivo* was due to the pancreatic microenvironment, we next transplanted the MPDAC-4 cell line subcutaneously and treated mice with tumours with CB-839. Similar to the orthotopic experiment, there was no significant tumour growth delay in mice bearing MPDAC-4 flank tumours (Fig. 3c, $P = 0.0957$, $t$-Test). We also examined the tumour growth of human PDAC flank xenografts and saw no significant tumour growth delay (Fig. 3d, $P = 0.7241$, $t$-test). To rule out inadequate drug delivery as a cause

of the lack of tumour growth delay, we measured CB-839 plasma drug levels at the end point of the MPDAC-4 flank tumour experiment and noted adequate CB-839 drug levels (Fig. 3e). Of note, we did not find any evidence of secondary tumours in pancreata in the orthotopic experiment nor development of metastatic tumour deposits in either the orthotopic or the flank tumour experiments suggesting that this phenomenon in the autochthonous model was due to the nature of the model, rather than a physiological situation. Together, these data support the idea that *in vivo* PDAC tumours do not respond to GLSi and this is not dependent on the location of where the tumour is grown.

**GLSi metabolic profiling identifies upregulated pathways.** One potential explanation for the ineffectiveness of CB-839 *in vivo* was an adaptive response to chronic exposure of GLSi. To model this scenario, we performed long-term proliferation assays *in vitro* with CB-839. Consistent with this hypothesis, PDAC lines re-established their baseline proliferative rate at later time points, even at higher concentrations of CB-839, suggesting some adaptive response (Fig. 4a, Supplementary Fig. 3a,b). Long-term treatment with BPTES revealed similar findings (Supplementary Fig. 3c). To determine the nature of this adaptive response, we examined relative metabolite pools in MPDAC-4 cells treated with CB-839 at various time-points (Fig. 4b, Supplementary Data 1). When examining metabolites immediately upstream and downstream of GLS, we noted that at 72 h the cells maintained a significant increase in the Gln levels as well as decreases in Asp and malate (Fig. 4c, $P < 0.05$, $t$-test). However, in contrast to the short-term

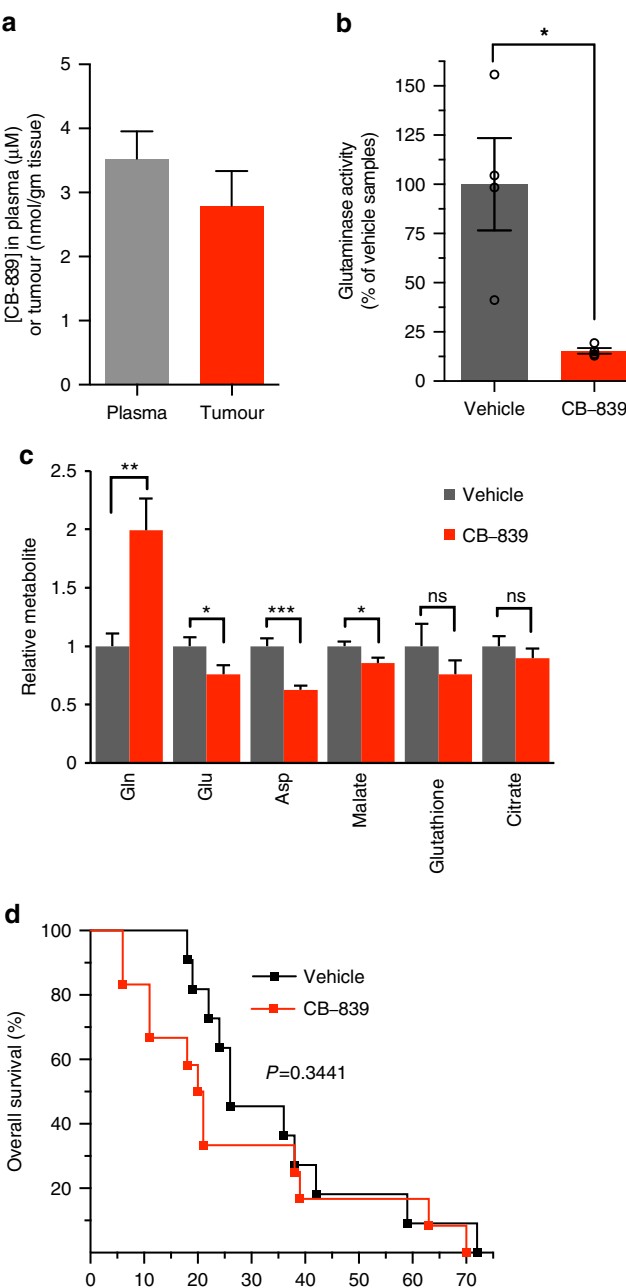

**Figure 2 | CB-839 treatment has no antitumour activity in an autochthonous mouse model of PDAC.** (**a**) CB-839 levels measured by LC/MS-MS in plasma and tumour samples 4 h after oral dosing of 200 mg kg$^{-1}$ CB-839 of LSL-Kras$^{G12D}$; p53 L/+, Pdx1-Cre mice bearing pancreatic tumours (plasma, $n = 5$ independent mice; tumour, two separate pieces from each tumour of five mice, $n = 10$). Data are represented as mean ± s.e.m. (**b**) Glutaminase activity measured in tumour lysates from animals ($n = 4$ per group) treated with vehicle or CB-839 as in (**a**). The percent inhibition by CB-839 relative to vehicle is plotted, $n = 4$. Data are represented as mean ± s.e.m. (**c**) Relative metabolite abundance in tumour lysates from animals treated as in (**a**) (2–3 tumour pieces from five mice per group, $n = 11$ vehicle treated, $n = 10$ CB-839 treated). Data are presented as mean total metabolite pools ± s.e.m. (**d**) Kaplan–Meier analysis comparing survival of vehicle (black, $n = 11$) and CB-839 (red, $n = 13$) treated LSL-Kras$^{G12D}$; p53 L/+, Pdx1-Cre mice bearing pancreatic tumours. There was no significant effect on survival ($P = 0.3441$ by Log-rank (Mantel–Cox) test). For panels **b,c**, significance determined by t-test, *$P < 0.05$, **$P < 0.01$, ***$P < 0.001$, ns: non-significant, $P > 0.05$.

treatment (Fig. 1h), these cells no longer displayed decreases in Glu, α-KG, GSH and succinate (Fig. 4c). Metabolite set enrichment analysis of upregulated and downregulated metabolites (Fig. 4d, Supplementary Fig. 3d, Supplementary Data 2) revealed an enrichment in protein biosynthesis, amino acid metabolism and oxidation of branched chain fatty acid metabolism terms in the upregulated metabolites set (Fig. 4d). Metabolites involved in the oxidation of branched chain fatty acid metabolism included propionylcarnitine, acetylcarnitine and C5-carnitines (2-Methylbutyr-oylcarnitine and isovalerylcarnitine) (Fig. 4e).

To understand better the adaptive changes in metabolic response, we employed uniformly labelled $^{13}$C-labelled Gln ([U-$^{13}$C$_5$]Gln) tracing to identify the itinerary of Gln-derived carbons in control versus CB-839 treated conditions. Short-term treatment (6 h) revealed decreases in the absolute amounts of most species of Glu as well as several other downstream metabolites (including Asp and malate) but relative increases in unlabelled Glu, indicating an increase in alternative pathways for Glu (Supplementary Fig. 3e,f, Supplementary Data 3). At later time points (72 h) Gln-tracing showed an increase in the relative and absolute amounts of unlabelled Glu as well as several other downstream metabolites (non-Gln derived) indicating an increase in alternative pathways for Glu and related metabolite production (Supplementary Fig. 3e,f, Supplementary Data 3). At the same time, an increase in Gln-derived Glu also suggested some Gln-dependent pathways (but not necessarily GLS-dependent) contributing to the increase back to baseline of Glu.

Metabolomic profiling of human PDAC cells revealed a similar enrichment in oxidation of branched chain fatty acid metabolism (Supplementary Fig. 3g, Fig. 4e). Interestingly, despite a similar re-establishment of proliferative capacity as mouse PDAC cells, human PDAC cells maintained significant decreases of Glu, Asp, α-KG, malate, GSH, succinate and citrate after 72 h of CB-839 treatment (Supplementary Fig. 3h, $P \leq 0.02$, for all, t-test) suggesting that the relevant *in vitro* timing of treatment or overall nature of the adaptive metabolomic response to GLSi may differ between individual pancreatic cancers. Together these data illustrate that depriving PDAC cells of their preferred carbon source for Glu leads to attempts by the cell to procure carbon from alternative pathways. Furthermore, the response to perturbation of metabolic pathways in cell culture may predict the metabolic pathways on which PDAC tumours are dependent on *in vivo*, when assessed over a longer period to allow for the adaptation that will occur *in vivo*. Indeed, metabolite analysis of mouse PDAC xenograft tumours and autochthonous tumours from mice treated long-term (>2 weeks) with vehicle or CB-839, revealed a continued increase in Gln in CB-839 treated tumours but no significant difference in core metabolites (Fig. 4f, $P \geq 0.1$, for all, t-test, Supplementary Fig. 3i). Consistent with the *in vitro* studies, the metabolites involved in the oxidation of branched chain fatty acids (Fig. 4g) were also elevated in CB-839 treated tumours (Fig. 4b,e). The early decrease in Glu and other metabolites, combined with the reaccumulation of these metabolites at a later time point as well as other specific changes observed (increase in fatty acid metabolism-associated carnitines), suggests some reliance on GLS-derived Glu that is rescued by an alternative metabolic pathway/pathways.

**GLSi quantitative proteomics reveals compensatory pathways.** To further determine the nature of the adaptive response, we used multiplexed isobaric tag-based quantitative mass spectrometry to analyze the proteomic response to CB-839 treatment[27]. We first compared the proteomes of untreated, CB-839 treated for 24 h and 72 h MPDAC-4 cells (Supplementary Fig. 4a, Supplementary Data 4). The magnitude of changes in the CB-839-24 h treatment

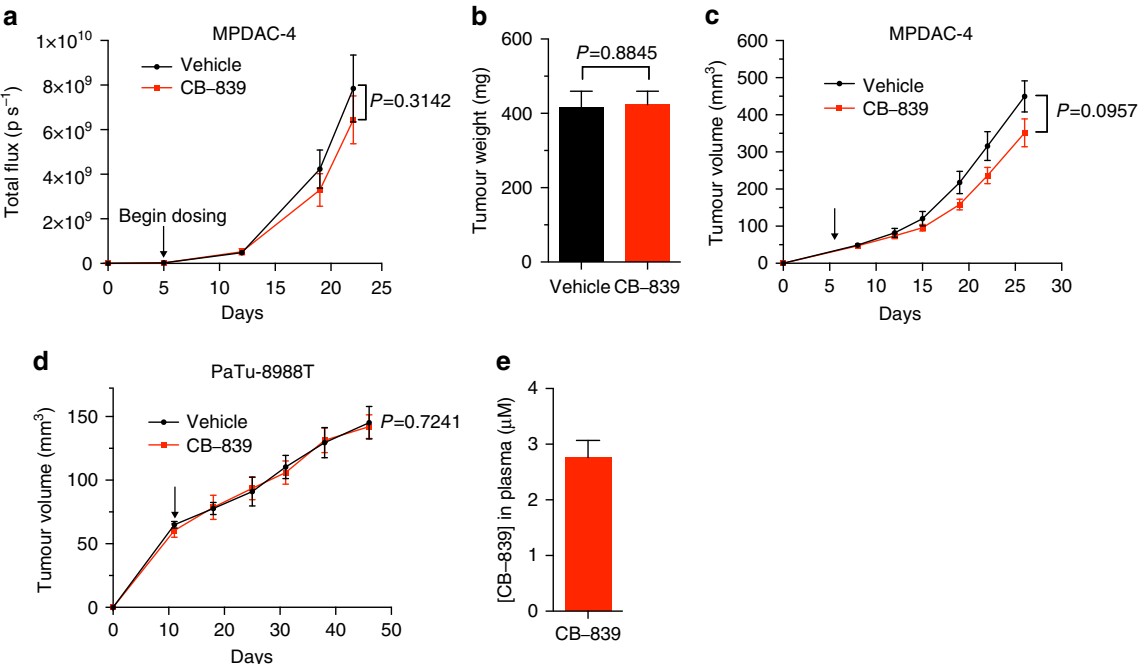

**Figure 3 | CB-839 treatment has no antitumour activity in cell line-derived transplanted mouse models of PDAC. (a)** MPDAC-4 cells constitutively expressing luciferase were implanted into the pancreata of nude mice. Mice were then randomized to CB-839 treatment (200 mg kg$^{-1}$, twice daily) or control (vehicle) treatment ($n = 13$ vehicle arm, $n = 16$ CB-839 arm) and imaged weekly with luciferase imaging. Arrow indicates treatment initiation. Data were expressed as mean total flux (p/s) ± s.e.m. A $t$-test was performed at the end point, $P = 0.3142$. **(b)** Tumour weight from mice at the end point of experiment in (**a**) presented as mean ± s.e.m. $P = 0.8845$ ($t$-test). **(c)** MPDAC-4 cells were implanted subcutaneously in nude mice and after the average tumour size of the cohort reached ∼50 mm$^3$ mice were randomized and treated as indicated with vehicle or CB-839 (200 mg kg$^{-1}$, twice daily), (vehicle, $n = 14$; CB-839, $n = 13$). Tumours were measured twice a week. Error bars represent ± s.e.m. A $t$-test was performed at the end point, $P = 0.0957$. **(d)** PaTu-8988T cells were grown as xenograft tumours in the flanks of nude mice, mice were randomized and treated as in (**c**). Tumours were measured weekly (vehicle, $n = 9$; CB-839, $n = 8$). Error bars represent ± s.e.m. A $t$-test was performed at the end point, $P = 0.7241$. **(e)** CB-839 levels in plasma from MPDAC-4 tumour bearing mice treated with CB-839 in (**c**), measured at end point, $n = 13$. Data presented as mean ± s.e.m.

data set was lower than the CB-839-72 h, likely owing to the earlier time-point and a lag in change in protein expression in response to GLSi as demonstrated by Principal component analysis (PCA) (Fig. 5a–c). As the majority of changes were identified at the 72 h time point, we employed gene set enrichment analysis (GSEA) comparing DMSO-treated to 72 h CB-839 treatment[28] (Supplementary Data 5, see methods) and analysed results using an enrichment map strategy[29] to determine the global response to CB-839 treatment (Fig. 5d). Enrichment map nodes associated with upregulated proteins included oxidative stress response, fatty acid and lipid metabolism, glycolysis, oxidative phosphorylation, amino acid metabolism and lysosomal processes (Fig. 5d, Supplementary Data 5). The marked enrichment for lipid and fatty acid-related processes was intriguing given the results from metabolomic analyses and is in line with Gln metabolism supporting fatty acid synthesis[30]. In addition, we observed increased expression of multiple proteins involved in the oxidative stress response (Fig. 5e, Supplementary Fig. 4b). We also measured proteome changes in a separate CB-839-sensitive cell line, PaTu-8988T (Supplementary Data 6). We saw similar changes with respect to upregulation of oxidative stress response proteins. Although there were not marked changes in all Nrf2 antioxidant response pathway targets, there was likely some participation of the Nrf2 pathway in this response (Supplementary Fig. 4c). In line with a decrease in proliferative capacity and stress response upon GLSi, GSEA terms associated with downregulated proteins clustered in cell cycle, RNA and DNA metabolism, transcription, translation and protein-folding nodes (Fig. 5d, Supplementary Data 5).

Given the reaccumulation of Glu in CB-839-treated cells, we were also interested in examining levels of Glu-producing enzymes. In MPDAC-4 cells at 72 h, GLS and GLS2, normally expressed in the liver, are not consistently up- or downregulated (Supplementary Fig. 4d, Supplementary Data 7). We also examined changes in Gln amidotransferases that could account for reaccumulation of fully labelled Glu (M + 5, Supplementary Fig. 3e). Among the Gln amidotransferases, asparagine synthetase (ASNS) was significantly increased after CB-839 treatment suggesting ASNS may contribute to Glu reaccumulation in the setting of GLSi (Supplementary Fig. 4e, $P = 5.56E-11$, $t$-test, Supplementary Data 7). Of note, ASNS has also been implicated in cell survival upon Gln withdrawal via multiple mechanisms[31,32]. Although not a measure of flux, the remaining Gln amidotransferases were close to baseline measurements (Supplementary Data 7). Other Gln-independent, Glu-producing enzymes were elevated in MPDAC-4 cells including branched chain aminotransferase 1 (BCAT1), alanine aminotransferase 2 (GPT2), Gamma-Glutamyl hydrolase (GGH) and 5-oxoprolinase (OPLAH), suggesting alternate pathways for Glu production not reliant on GLS (Supplementary Fig. 4f, Supplementary Data 7).

Taken together, the metabolomic and proteomic response to GLSi reflects the complex number and variety of cellular processes in which Gln metabolism is involved[30,33]. Indeed, an integrated metabolomic and proteomic analysis confirmed the importance of both GSH and fatty acid metabolism as well as a number of additional metabolic pathways in the response to GLSi (Supplementary Fig. 5a, Supplementary Data 8)[34].

**GLSi proteomics predicts rational combinatorial treatments.** Based on metabolomics and proteomic changes, we were interested whether rational combinatorial approaches could lead to synergy or sustained treatment response. We were first

interested in the upregulation of oxidative stress response proteins given prior data showing the importance of Gln metabolism in PDAC to support redox homoeostasis[9] as well as synergy with NADPH-depleting agents[35]. As previously

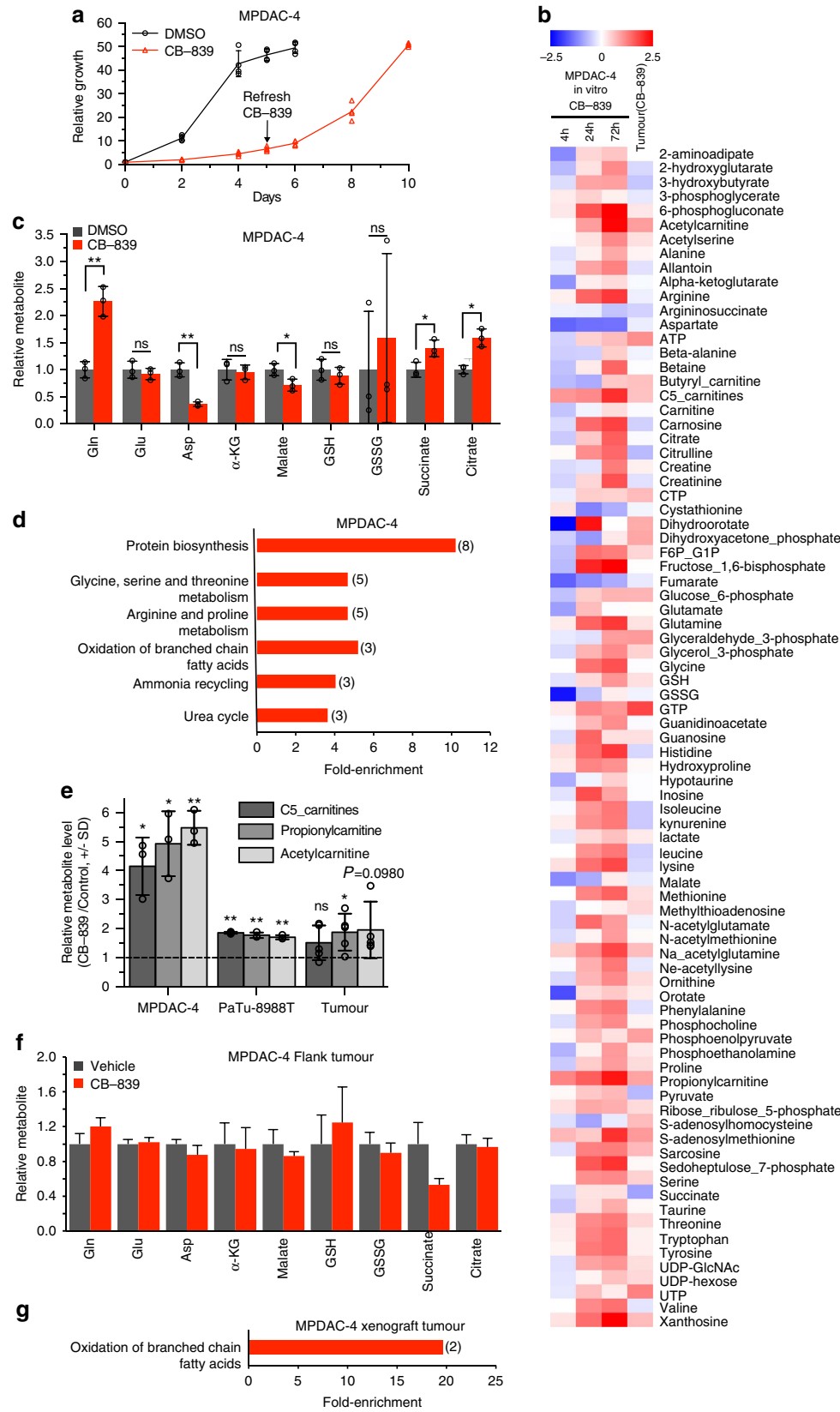

demonstrated, PDAC cells inhibited for glutaminolysis had significant increases in ROS at early time points after GLSi (Fig. 6a, $P = 0.02329$, $t$-test, Supplementary Fig. 5b, $P = 0.00208$, $t$-test). However, at long-term treatment and in CB-839-resistant cells (CB-839-R: cells treated with CB-839 for greater than 15 days that resumed proliferation), ROS levels decreased close to baseline suggestive of an adaptive response to treatment (Fig. 6a, Supplementary Fig. 5b). In addition, GSH levels initially decreased significantly as did the reduced to oxidized ratio and these returned to baseline in CB-839-R cells (Fig. 6b, $P \leq 0.003$, for all, $t$-test). In line with these results, N-acetyl cysteine partially rescued cell proliferation when added at the outset of CB-839 inhibition and hydrogen peroxide added to CB-839 produced a synergistic response (Fig. 6c, d). Based on the sustained elevation of proteins involved in anti-oxidant response (CTH and ALDH1L2, Supplementary Fig. 5c), we predicted that targeting the related redox pathways in combination with CB-389 may produce a synergistic response. Given the lack of potent inhibitors of CTH, we elected to use L-Buthionine-(S,R)-sulfoximine (BSO), an inhibitor of γ-glutamylcysteine (GCLC/GCLM), which is downstream of CTH and important for GSH synthesis (Fig. 5e)[11,36]. We tested the combination of CB-839 and BSO both at initial time points and in CB-839 resistant cells. Although single-agent BSO had no effect on cells, co-treatment abrogated the regrowth of CB-839 treated cells (Fig. 7a, Supplementary Fig. 5d). In addition, combinatorial BSO and CB-839 treatment in CB-839-R cells also decreased proliferation (Fig. 7b). Likewise, we tested inhibition of ALDH1L2-related folate pathways important for NADPH production with methotrexate, an inhibitor of a dihydrofolate reductase. Combination therapy showed an additive effect of methotrexate and a blunting of the CB-839 resistance phenotype (Fig. 7c,d).

Given the marked increase in fatty acid oxidation-related metabolites and proteome changes suggestive of an increase in fatty acid metabolism in response to CB-839, we also tested the combination of CB-839 and etomoxir, a carnitine palmitotyl-transferase I inhibitor that prevents shuttling of fatty acids to the mitochondria for oxidation[37]. Similar to BSO treatment, etomoxir had no effect as a single agent but decreased the regrowth of CB-839-treated cells when used concurrently (Fig. 7e). In contrast to BSO, etomoxir treatment in CB-839-R cells was ineffective, suggesting that the increased reliance on fatty acid metabolism was transitory during a shifting adaptive response (see below) (Fig. 7f).

Based on positive results of BSO combination with GLSi in vitro, we were interested if combinatorial targeting in vivo would prove useful. We therefore treated mice transplanted with MPDAC-4 cells with fully formed tumours with CB-839 with BSO. Strikingly, combination therapy resulted in significant tumour growth inhibition (Fig. 7g,h, $P = 0.008237$, $t$-test). We also attempted a triple combination therapy with GLSi, BSO and etomoxir given positive results in cell culture; however, the dual combination of CB-839 and etomoxir was lethal in all mouse models tested (see methods). Taken together, these results indicate that rational combinatorial targeting of adaptive proliferation/metabolic pathways upregulated in response to GLSi can be an effective strategy for tumour control but that normal tissue toxicity must be carefully assessed as well.

**GLSi network analysis identifies combinatorial treatments.** The shift in sensitivity to fatty acid oxidation inhibitors suggested that in addition to proteomic evaluation at early time points in the acute response to GLSi, evaluating the proteomic response at long time points when cells have regained proliferative capacity could be informative. We therefore profiled MPDAC-4 and PaTu-8988T cells resistant to GLSi (Fig. 8a, Supplementary Data 6, 9). Quantitative proteomics of CB-839-R cells revealed a number of sustained proteomic changes including in oxidative stress response proteins (Supplementary Fig. 6a, Supplementary Data 7) as well as enzymes previously shown to be involved in Gln -deprivation cell survival stress response (ASNS and pyruvate carboxylase in MPDAC-4 cells)[31,32,38,39]. Notably, enzymes related to fatty acid metabolism were no longer enriched, which aligns with a decreased sensitivity to etomoxir (Supplementary Fig. 6b, Supplementary Data 7). GSEA paired with enrichment map analysis revealed upregulated nodes for oxidative stress response, amino acid metabolism, lysosomal processes, glycolysis and pyruvate metabolism (Fig. 8b, Supplementary Data 10). Despite a return to baseline proliferation, downregulated nodes included mRNA metabolism, transcription, translation, as well as extracellular matrix regulation (Fig. 8b).

We were also interested in examining the GSEA pattern comparing CB-839-72 h treated cells to CB-839-R cells to understand what processes active at 72 h may have reversed or were maintained. Many of the pathways enriched at 72 h in comparison with untreated cells (Fig. 5d), were downregulated in resistant cells in comparison with the 72 h time point, including fatty acid, amino acid, and pyruvate metabolism, lysosomal processes and the tricarboxylic acid cycle (Fig. 8c, Supplementary Data 10). Conversely, many of the processes downregulated at 72 h (Fig. 5d) were enriched in resistant cells in comparison with 72 h including DNA and RNA metabolism, transcription, translation and protein folding (Fig. 8c). Overall, this suggests that there was a return towards the basal/untreated state as reflected in the PCA analysis (Fig. 8a). However, as noted in Fig. 8b, several of the pathways downregulated in resistant cells in comparison with the CB-839-72 h time point, including pyruvate

**Figure 4 | PDAC GLSi metabolic profiling identifies common upregulated pathways.** (**a**) Relative proliferation of MPDAC-4 cell line treated long-term with CB-839 (1 μm) or DMSO. As indicated CB-839 was refreshed at the mid-point of the experiment. Error bars depict ± s.d. of four independent wells from a representative experiment (of four experiments). (**b**) Log$_2$(CB-839 treated sample/control) heatmap for metabolites measured from MPDAC-4 cell culture (at 4, 24 and 72 h) and MPDAC-4 xenograft tumour experiment from Fig. 3c. Values presented for cell culture data are the mean of three independent wells from a representative experiment (of three experiments). Values presented for tumour are the mean of five independent tumours. (**c**) Relative metabolite abundance in MPDAC-4 cells following 72 h CB-839 treatment. Data are presented as means of total metabolite pools ± s.d. of three independent wells from a representative experiment (of three experiments). (**d**) Metabolite set enrichment analysis (MSEA) of upregulated metabolites from MPDAC-4 72 h CB-839 metabolomics experiment as performed in (**b**), statistically significant terms are graphed according to fold-enrichment (number of metabolites represented in term in parentheses). (**e**) Relative metabolite levels (CB-839 treated sample/control) plotted for carnitine metabolites from cell culture experiments (MPDAC-4, PaTu-8988T, means of total metabolite pools ± s.d. of three independent wells from a representative experiment (of three experiments)) and MPDAC-4 tumour experiment (tumours from experiment described in Fig. 3c, means of total metabolite pools ± s.d., $n = 5$ independent tumours). (**f**) Relative metabolite abundance in MPDAC-4 flank tumours as in Fig. 3c following long-term CB-839 treatment, $n = 5$ tumours per group, error bars represent s.e.m. All comparisons non-significant, $P > 0.05$. (**g**) Metabolite set enrichment analysis (MSEA) of significantly upregulated metabolites from MPDAC-4 tumour CB-839 metabolomics experiment. For **c,e** and **f**, significance determined by $t$-test, *$P < 0.05$, **$P < 0.01$, ***$P < 0.001$, ns: non-significant, $P > 0.05$.

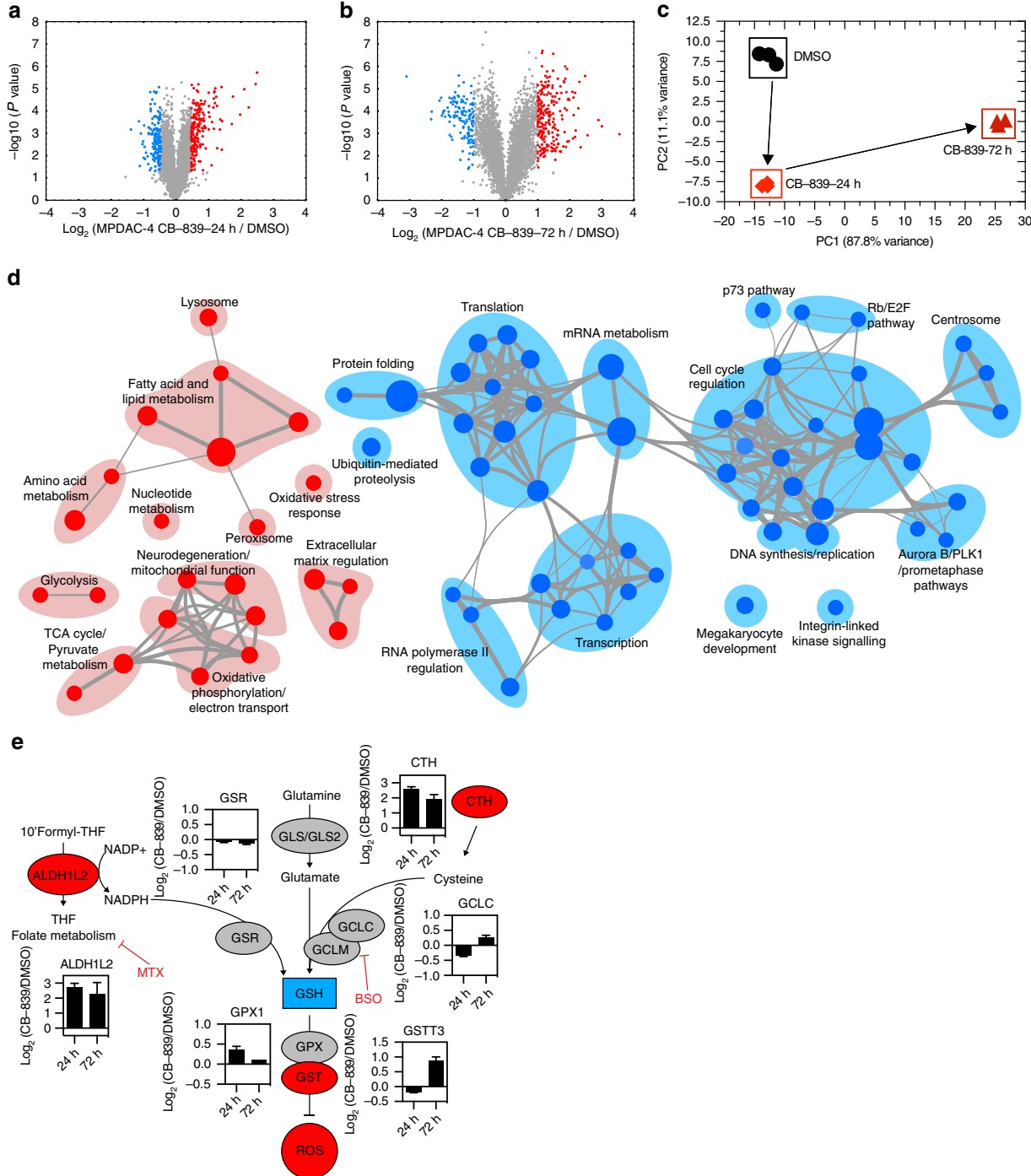

**Figure 5 | PDAC GLSi proteomics identifies up- and downregulated pathways.** (**a,b**) Volcano plots illustrate statistically significant protein abundance differences in MPDAC-4 cells treated with CB-839 for 24 h (**a**) or 72 h (**b**). Volcano plots display the $-\log_{10}(P\ value)$ versus the $\log_2$ of the relative protein abundance of (**a**) mean CB-839 24 h treatment or (**b**) mean CB-839 72 h treatment to mean control (DMSO). Red circles represent the top 5% upregulated proteins with a $P$ value $< 0.05$, blue circles represent the 5% most downregulated proteins and a $P$ value $< 0.05$. The remainder of proteins are represented as gray circles. Data are from three independent plates from a representative experiment of three experiments. (**c**) Principal component analysis of the CB-839 treated proteomes represented in a two-dimensional space. (**d**) Enrichment map of gene set enrichment analysis (GSEA) of CB-839-72 h versus DMSO, FDR $< 0.01$, co-efficient overlap (CO) $> 0.25$, node size is related to number of components identified within a gene set and the width of the line is proportional to the overlap between related gene sets. GSEA terms associated with upregulated (red) and downregulated (blue) proteins are coloured accordingly and grouped into nodes with associated terms. (**e**) Proteins in the oxidative stress response protein network are upregulated in response to CB-839 treatment. Graphs represent mean of $\log_2$(CB-839 treated sample/control) from MPDAC-4 Experiments 1–3 of select proteins in oxidative stress response pathways ($n = 9$, three independent plates for each Experiment 1–3). Error bars represent $\pm$ s.e.m. Upregulated (red), downregulated (blue), no relative change (grey). Abbreviations: ALDH1L2: Aldehyde Dehydrogenase 1 Family Member L2, GSR: Glutathione reductase, mitochondrial, GLS: Glutaminase, GLS2: Glutaminase 2, CTH: Cystathionine gamma-lyase, GCLC: Glutamate-cysteine ligase catalytic subunit, GCLM: Glutamate-cysteine ligase regulatory subunit, GPX1: Glutathione peroxidase 1, GSTT3: Glutathione S-transferase Theta 3, MTX: Methotrexate, BSO: L-Buthionine-*S,R*-sulfoximine.

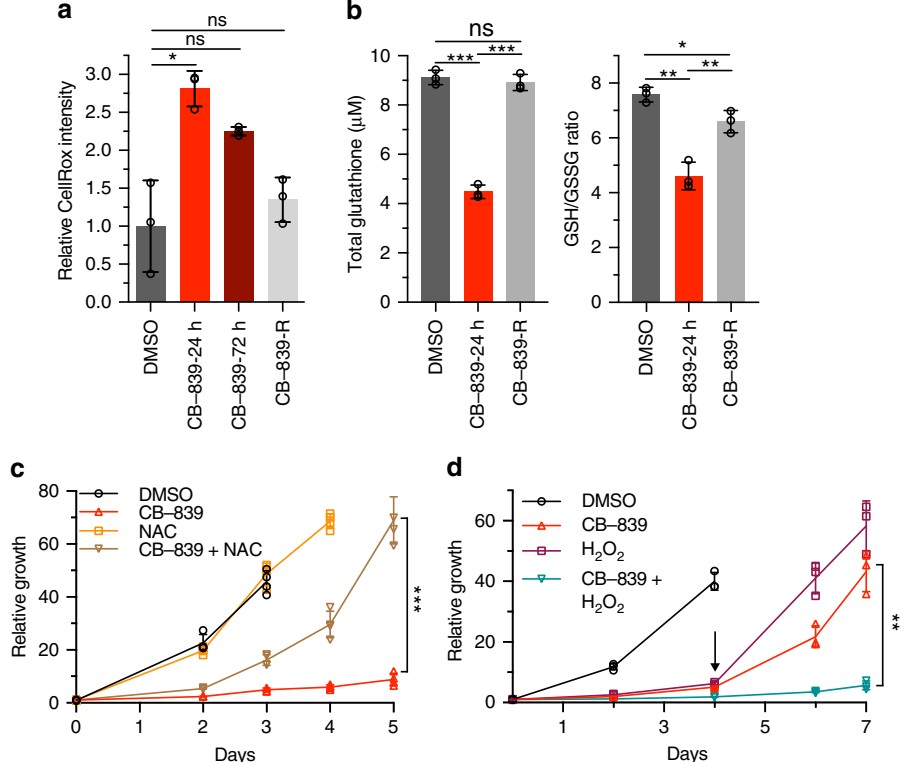

**Figure 6 | GLSi induces an oxidative stress response in PDAC cells. (a)** Relative CellRox intensity in MPDAC-4 cells following CB-839 treatment at the indicated times as measured by flow cytometry. Data are presented as mean ± s.d. of three independent wells from a representative experiment (of three experiments). **(b)** Total glutathione levels (left) and ratio of reduced to oxidized (right) after CB-839 treatment. Data are presented as mean ± s.d. of three independent wells from a representative experiment (of three experiments). **(c)** Relative proliferation of MPDAC-4 cell line treated with CB-839 or DMSO with or without *N*-acetyl cysteine (2 mM). Data are plotted as mean relative cell proliferation, error bars depict ± s.d. of four independent wells from a representative experiment (of three experiments). **(d)** Relative proliferation of MPDAC-4 cell line treated with CB-839 or DMSO with or without hydrogen peroxide ($H_2O_2$) (20 μm). Data are plotted as mean relative cell proliferation, error bars depict ± s.d. of three independent wells from a representative experiment (of three experiments). Arrow represents time point when treatments were refreshed. Significance determined by *t*-test in all panels. *$P < 0.05$, **$P < 0.01$, ***$P < 0.001$, ns: non-significant, $P > 0.05$.

and amino acid metabolism as well as lysosomal processes, were still upregulated in comparison with untreated samples consistent with the PCA analysis that showed resistant cells were still distinct from untreated cells (Fig. 8a,b). As expected, nodes associated with oxidative stress response were not increased or decreased in resistant cells in comparison with 72 h treated cells and as in Fig. 8b, remained elevated in resistant cells in comparison with untreated cells. This aligns with our data showing continued sensitivity of CB-839-R cells to oxidant-inducing drugs and provides further support for combinatorial targeting.

To advance our understanding of the functional responses to GLSi and identify synergistic drug combinations not predicted by GSEA or on a protein-by-protein basis, we paired our proteomic analysis with the Connectivity Map 2.0 database[40], which contains expression profiles in response to 1309 different compounds (Fig. 8d). Here, we identified drugs with responses that were positively or negatively correlated with CB-839 treatment at 24 and 72 h as well as in CB-839-R cells (Fig. 8e–h, Supplementary Data 11). In the CB-839-24 h versus dimethylsulphoxide (DMSO) comparisons, drugs positively correlated with the observed proteomic response included thapsigargin and withaferin-A (endoplasmic reticulum (ER) stress inducers), gossypol (LDHA inhibitor which induces oxidative stress), 15Δ-prostaglandin J2 (PPARγ inducer, a regulator of lipid metabolism), Prestwick-675 (albendazole: microtubule polymerization inhibitor), thioridazine (anti-psychotic) and MG-262 (proteasome inhibitor). At 72 h

positively correlated drugs included HDAC inhibitors (trichostatin, vorinostat), PI3-Kinase/mTOR inhibitors (LY-294002, wortmannin, quinostatin), alsterpaullone (CDK inhibitor), methotrexate and a topoisomerase inhibitor (camptothecin). Cyclosporine was positively correlated with changes in CB-839-R cells. Several of these classes of drugs have recently been shown to be associated with a synergistic response in combination with GLSi including HSP90 inhibition/ER stress activation and mTOR inhibition[41,42]. Interestingly, the Connectivity Map analysis revealed pathways suggested by but not highlighted by GSEA, including an induction of the ER stress response (thapsigargin and withaferin-A). An induction of the ER stress response was confirmed by RT-qPCR analysis of CHOP, an ER stress-induced transcription factor, and BiP, an ER chaperone protein (Supplementary Fig. 6c). On further analysis of upregulated MPDAC-4 proteins at 72 h after CB-839 treatment, a majority of them have been previously reported to be induced under conditions of ER stress (VLDLR, PYCR1, CTH, PCK2, ALDH1L2, AVIL, LONP1, Supplementary Data 4). As in Fig. 8c, we were also interested if the Connectivity Map analysis could identify pathways that were no longer active in CB-839-R cells as opposed to acutely treated cells (CB-839-72 h). Connectivity Map analysis of CB-839-R versus CB-839-72 h changes (Fig. 8h) revealed that a number of drugs previously positively correlated with changes at 72 h (Fig. 8f), were inversely correlated including, PI3-Kinase/mTOR inhibitors (Wortmannin, Ly-294002, Sirolimus), MS-275 (HDAC inhibitor),

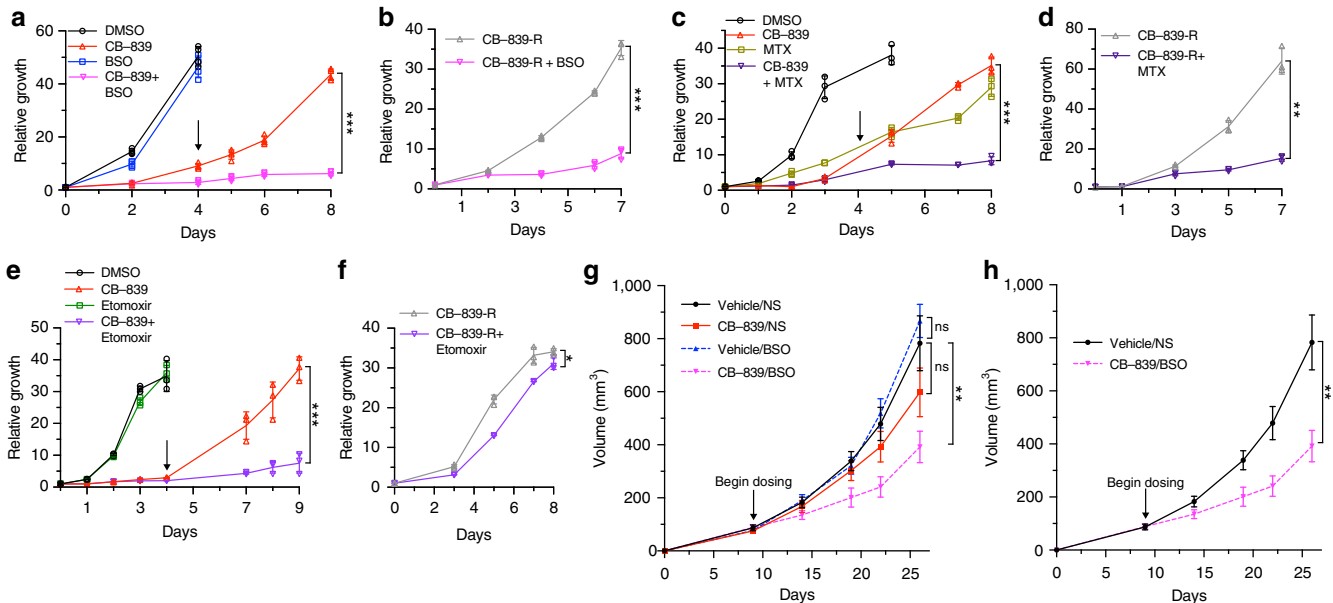

**Figure 7 | GLSi proteomics and metabolomics predicts responsiveness to oxidative stress targeted combination treatment.** (**a**) Relative proliferation of MPDAC-4 cell line treated with CB-839 or DMSO with or without BSO (100 μm). Data are plotted as mean relative cell proliferation, error bars depict ± s.d. of four independent wells from a representative experiment (of three experiments). Arrow represents time point when treatments were refreshed, similarly in **c**,**e**. (**b**) Relative proliferation of CB-839-resistant (CB-839-R) MPDAC-4 cell line treated with CB-839 alone or in combination with BSO (100 μm). Data are plotted as mean relative cell proliferation, error bars depict ± s.d. of three independent wells from a representative experiment (of three experiments). (**c**) Relative proliferation of MPDAC-4 cell line treated with CB-839 or DMSO with or without methotrexate (MTX) (50 nm). Data are plotted as mean relative cell proliferation, error bars depict ± s.d. of three independent wells from a representative experiment (of three experiments). (**d**) Relative proliferation of CB-839-R MPDAC-4 cell line treated with CB-839 alone or in combination with methotrexate (50 nm). Data are plotted as mean relative cell proliferation, error bars depict ± s.d. of three independent wells from a representative experiment (of three experiments). (**e**) Relative proliferation of MPDAC-4 cell line treated with CB-839 or DMSO with or without etomoxir (100 μm). Data are plotted as mean relative cell proliferation, error bars depict ± s.d. of three independent wells from a representative experiment (of three experiments). (**f**) Relative proliferation of CB-839-R MPDAC-4 cell line treated with CB-839 alone or in combination with etomoxir (100 μm). Data are plotted as mean relative cell proliferation, error bars depict ± s.d. of three independent wells from a representative experiment (of three experiments). (**g**) MPDAC-4 cells were implanted subcutaneously in nude mice and after the average tumour size of the cohort reached ∼50 mm³ mice were randomized and treated as indicated with vehicle, CB-839, BSO, or CB-839 and BSO, (n = 8 per arm), see methods for full details of dosing. Tumours were measured twice a week. Error bars represent ± s.e.m. (**h**) Data as in (**g**) reproduced without CB-839 and BSO arms. Error bars represent ± s.e.m. Significance determined by t-test in all panels. *P < 0.05, **P < 0.01, ***P < 0.001, ns: non-significant, P > 0.05.

and Tanespimycin (ER stress inducer). The reversal of the ER stress induction profile was recapitulated with RT-qPCR showing the CHOP and BiP levels in CB-839-R cells decreased from the peak CB-839-72 h values (Supplementary Fig. 6c).

To determine whether combination of GLSi and a select number of these drugs elicits a synergistic response, we tested several combinations in cell culture. Combination treatment with 17-AAG, albendazole and MG-132 elicited an additive response (Fig. 8i–k). As suggested by GSEA and Connectivity Map analysis, 17-AAG, albendazole and MG-132 were no longer as effective in CB-839-resistant cells (Supplementary Fig. 6d,e,f) highlighting the importance of timing of combination therapy administration.

## Discussion

Our results reveal that PDAC cells have adaptive metabolic networks that allow them to utilize available nutrients to sustain proliferation. Using GLSi as an example, we show that although the *in vitro* effects on proliferation were quite marked, these were lost in multiple *in vivo* models of PDAC. Through an integrated analysis of proteomics and metabolomics, we identify multiple compensatory pathways that may explain the resistance to GLSi and show as proof of concept that combining inhibitors

to these pathways with GLS inhibitors may have therapeutic utility. In addition to our study, several recent studies in PDAC as well as additional cancers have also identified combination therapies that target adaptive metabolic mechanisms of resistance or take advantage of adaptive responses to enhance therapy, albeit with less-comprehensive approaches[35,43].

The acute response to GLSi (Supplementary Fig. 7) is marked by induction of multiple stress response pathways including the ER stress response and anti-oxidant stress response. As a result of these and other responses, DNA synthesis, transcription, translation and protein folding are attenuated precipitating the observed decrease in proliferation. Another major area of acute adaptation is in re-wiring cellular metabolism. Alterations in metabolic enzymes, including increased expression of pyruvate carboxylase, can provide carbon to the tricarboxylic acid cycle via conversion of pyruvate to oxaloacetate and has been shown to be important for GLS-independent cell growth[38,39]. Activation of lipid biosynthetic pathways, in part via PPARγ signaling, supports findings that Gln is an important source for accumulation of fatty acids and that alternative pathways are necessary in response to GLSi acutely[30]. Nucleotide biosynthesis reactions are affected by GLSi contributing to a decrease in DNA synthesis. Amino acid metabolism is likewise affected acutely by GLSi likely given a decrease in Glu available for transamination

reactions. Finally, multiple enzymes capable of providing Glu via Gln-dependent and Gln-independent processes are upregulated all potentially contributing to replenishing Glu levels in MPDAC-4 cells. This final adaptation in the acute phase is likely responsible for MPDAC-4 cells regaining proliferation. However, PaTu-8988T cells never reestablish Glu levels yet are able to continue proliferation so they likely utilize alternative pathways that may not be Gln-dependent. CB-839-R cells

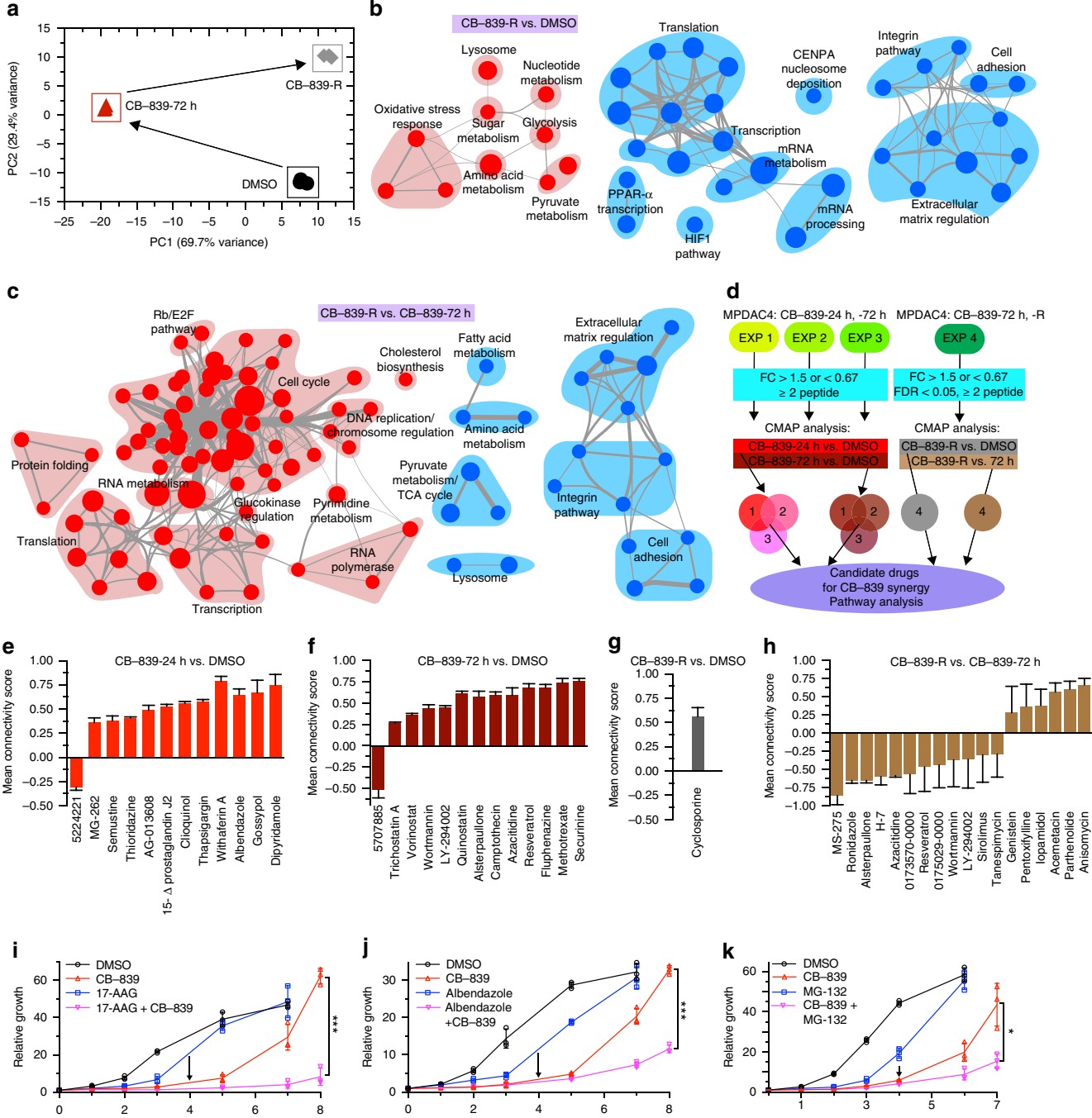

**Figure 8 | GLSi proteomics and metabolomics analysis predicts responsiveness to combinatorial treatment.** (**a**) Principal component analysis of the CB-839 treated proteomes represented in a two-dimensional space. (**b**) Enrichment map of GSEA of CB-839-R versus DMSO, FDR < 0.05, co-efficient overlap (CO) > 0.1, otherwise plotted as in Fig. 5d. (**c**) Enrichment map of GSEA of CB-839-R versus CB-839-72 h, FDR < 0.01, co-efficient overlap (CO) > 0.25. (**d**) Connectivity map (CMAP) workflow schematic for identification of candidate drugs for CB-839 synergy and pathway analysis. (**e**) CMAP analysis CB-839-24 h versus DMSO. Data represent mean connectivity score ± s.d. as determined from CMAP analysis of Experiments 1–3 independently. (**f**) CMAP analysis CB-839-72 h versus DMSO as in (**e**). (**g**) CMAP analysis CB-839-R versus DMSO. Data represent mean connectivity score ± s.d. from CMAP analysis of Experiment 4. (**h**) CMAP analysis CB-839-R versus CB-839-72 h as in **g**. (**i**) Relative proliferation of MPDAC-4 cell line treated with CB-839 or DMSO with or without 17-AAG (tanespimycin) (500 nm). (**j**) Relative proliferation of MPDAC-4 cell line treated with CB-839 or DMSO with or without albendazole (Prestwick-675) (400 nm). (**k**) Relative proliferation of MPDAC-4 cell line treated with CB-839 or DMSO with or without MG-132 (500 nm). Data for **i,j,k** are plotted as mean relative cell proliferation, error bars depict ± s.d. of three independent wells from a representative experiment (of three experiments). For panels **i,j,k**, significance determined by t-test, *P < 0.05, **P < 0.01, ***P < 0.001, ns: non-significant, P > 0.05.

maintain an elevated oxidative stress response, an increase in lysosomal processes, and upregulated glycolysis, nucleotide, sugar, amino acid and pyruvate metabolism. These resistant cells also appear to operate at a new basal level of ER stress and as such upregulate protein folding capacity to compensate for proteotoxic stress in comparison with acutely treated cells. It will be informative to compare proteomic responses across additional PDAC cell lines as well as additional GLSi-sensitive and insensitive cancers to understand what are the conserved proteomic responses that may direct combination therapy.

Several additional important findings emerged from our studies. First, how one performs the *in vitro* assessments of efficacy is extremely important. Standard 2 or 3-day growth assays may not be appropriate for inhibitors of cellular metabolism. Indeed, we noted upon longer term assays, that the cells began to regrow, which mirrored the lack of effect seen in the *in vivo* studies. Attempts at better modeling the *in vivo* growth conditions are warranted to faithfully model treatment responses *in vitro*. This includes more physiological media[44], oxygen tension[45] and potentially organoid growth conditions[46].

Second, although we were unable to identify a specific cause of the CB-839-induced secondary tumour formation in our genetically engineered mouse model of PDAC, further investigation may reveal important metabolic factors in determining PDAC tumour initiation. We currently speculate that given the importance of redox homoeostasis/ROS in generation of tumours[11,12,47], that GLSi may cause a change in the redox environment of pre-invasive or early invasive oncogenic Kras expressing pancreatic cells in this model thereby promoting tumour initiation/progression. These hypotheses require additional testing, but the observed phenotypes are likely a consequence of the limitations of the genetically engineered PDAC mouse model.

Finally, integrating proteomic analyses with metabolomics can be extremely useful in determining which metabolic adaptations to GLSi might be critical to sustained growth. Determining metabolic fluxes *in vivo* is quite difficult and the presence of stroma will confound such analyses[10]. Because proteins are relatively stable, sorting or microdissection of the tumour cells is certainly feasible, whereas the labile nature of metabolic reactions likely precludes such manipulations. Therefore, a proteomics approach to determine the metabolic adaptations in a particular patient's tumour may allow for the tailoring of patient specific metabolic inhibitor cocktails. Our data, even in a limited number of PDAC lines, suggest that individual tumours may have unique kinetics or metabolic adaptations to the same metabolic perturbations. Protein expression is also readily adaptable to clinical scenarios, where if one could define metabolic biomarkers these could be assessed by immunohistochemistry (IHC) on tumour biopsies.

In conclusion, pancreatic cancers have striking metabolic plasticity. Carefully executed *in vitro* analyses can help understand the metabolic adaptations to metabolic perturbations. By understanding the modalities of adaptation, we can develop novel and more effective therapeutic targets. Indeed, our analyses yielded several interesting combinatorial approaches that showed efficacy. Although combination metabolic inhibition may be a promising strategy, identifying whether these same synergies exist in normal tissues will be critical to ensure a therapeutic index. Further studies into the mechanisms of resistance and metabolic adaptation will help create more promising therapeutic options.

## Methods

**Cell culture.** IMR90, PANC-1 and BxPC-3 cell lines were obtained from the American Type Culture Collection. PaTu-8988t and PaTu-8902 that were obtained from the German Collection of Microorganisms and Cell Cultures. Primary mouse PDAC lines (MPDAC-1 and MPDAC-4) were generated from tumours from

genetically engineered mice (LSL-Kras$^{G12D}$; p53 L/+, Pdx1-Cre)[24], mouse embryonic fibroblasts were generated from a LSL-Kras$^{G12D}$; p53 L/+ mouse. Cell lines were authenticated by fingerprinting as well as visual inspection and carefully maintained in a centralized cell bank. All cell lines were tested routinely, and before all metabolomics and proteomic analyses, for mycoplasma contamination. Cell lines were cultured in DMEM (Invitrogen 11965) with 10% fetal bovine serum (FBS) and 1% Penicillin/Streptomycin except for Supplementary Fig. 1i where cells were cultured in RPMI-1640 (Invitrogen 11875) supplemented with 10% FBS and 1% penicillin/streptomycin and in Supplementary Fig. 1j where cells were grown on ultra-low attachment plates (Sigma, CLS7007) in 2% growth factor reduced matrigel (BD, 354230) supplemented with DMEM with 10% FBS and 1% penicillin/streptomycin.

**Cell proliferation assay.** Cells were plated in 24-well plates at 2,000–7,000 cells (depending on cell line) per well in 0.5 ml of media. The day after plating, cells were treated with CB-839 (100 nm unless otherwise indicated). For long-term assays (>5 days), drugs were refreshed at the mid-point of the experiment. At the indicated time points, cells were fixed in 10% formalin and stained with 0.1% crystal violet. Dye was extracted with 10% acetic acid and the relative proliferation was determined by OD at 595 nm.

**IC50 assay.** Cells were plated in 96-well plates and treated by serial dilution of CB-839 the day after plating for 48–72 h. Cell viability was measured using the CellTiter-Glo assay (Promega, G7570) according to the manufacturer's instructions. The IC50 was calculated using Graphpad Prism using three-parameter nonlinear regression analysis.

**Trypan assay.** Cells were treated in biological triplicate with DMSO or the indicated concentrations of drug including Staurosporine (Sigma, S5921). After 72 h, percent survival was assessed using a trypan blue exclusion assay.

**Mouse treatment studies and ultrasound.** Cohorts of genetically engineered mice were generated from a single colony (LSL-Kras$^{G12D}$; p53 L/+; Pdx1-Cre) under DFCI IACUC protocol 10-055. Beginning at 8 weeks of age, mice were monitored for tumour formation via abdominal palpation[26]. Tumour identification was confirmed and dimensions and volume were measured using high-resolution ultrasound (Vevo 770). In brief, mice were anesthetized using 1.5% isoflurane, abdominal fur was removed using fine clippers and depilatory cream. Pre-warmed sterile saline (1–2 ml) was administered via intraperitoneal (IP) injection. Ultrasound gel was applied over the entire abdominal area and the ultrasound transducer was used to identify abdominal landmark organs (liver/spleen) followed by the pancreas and the tumour. Once identified, the transducer was transferred to the 3D motor stage and a 3D scan was performed for measurement of tumour dimensions and volume. Tumour volumes were contoured for serial volumetric measurements[26]. For treatment studies, mice with tumours in the 50–100 mm$^3$ size range were enroled in a randomized fashion to either vehicle or CB-839 (vehicle, $n = 11$, CB-839, $n = 13$, average age: 24 weeks, mixed male and female, LSL-Kras$^{G12D}$; p53 L/+; Pdx1-Cre, mixed background (FVB and C57-Bl/6j)). Enrollment on the trial was balanced between arms in terms of starting tumour size, location of tumour within the pancreas, age of mice, and female versus male distribution. The vehicle consisted of 25% (weight/volume) hydroxypropyl-β-cyclodextrin (Roquette) in 10 mmol/l citrate, pH 2. CB-839 was formulated as a solution at 20 mg ml$^{-1}$ (w/v) in vehicle. Mice were dosed twice daily via oral gavage with 200 mg kg$^{-1}$ CB-839 or vehicle[15]. Daily treatment was continued until end point criteria were met. End point criteria included the development of haemorrhagic abdominal ascites, severe cachexia, weight loss exceeding 15% of initial weight, or extreme weakness or inactivity, in accordance with DFCI 10-055 IACUC approved protocol. Kaplan–Meier curves were plotted for survival studies and a log-rank test was used to compare the differences in survival probabilities. For pharmacokinetic and pharmacodynamics studies, mice were dosed and tumours and plasma were harvested at time points as specified. For orthotopic tumours $1 \times 10^4$ MPDAC-4 cells expressing luciferase were implanted into the tail of the pancreas of 8-week-old female NCr nude mice (Taconic) in 1:1 Matrigel (vehicle, $n = 13$, CB-839, $n = 16$). Mice were randomized to receive CB-839 or vehicle daily (as above), beginning 7 days after injection and luciferase imaging was performed weekly at the DFCI Small Animal Imaging Facility. Tumours were weighed at time of sacrifice. For subcutaneous xenografts, MPDAC-4 ($1 \times 10^6$) or PATU-8988T ($5 \times 10^6$ cells) cells, suspended in matrigel were injected subcutaneously into the lower flank of 8-week-old female NCr nude mice (Taconic) (MPDAC-4, vehicle, $n = 14$, CB-839, $n = 13$) (PaTu-8988T, vehicle, $n = 9$, CB-839, $n = 8$). Mice were randomized to receive CB-839 or vehicle daily when the average tumour volume was 50 mm$^3$. Tumour length and width were measured weekly and the volume was calculated according to the formula: $V = (\text{width}^2 \times \text{length})/2$. For combination studies with BSO, MPDAC-4 subcutaneous xenografts were generated as above in 8-week-old female NCr nude mice and were randomized to one of four treatment arms: (1) Vehicle oral gavage twice daily/Normal saline IP injection once daily, (2) CB-839 200 mg kg$^{-1}$ oral gavage twice daily/normal saline IP injection once daily, (3) Vehicle oral gavage twice daily/BSO 400 mg kg$^{-1}$ once daily and (4) CB-839 200 mg kg$^{-1}$ oral gavage

twice daily/BSO 400 mg kg$^{-1}$ once daily ($n=8$ mice per arm). For triple combination trials with CB-839, BSO and etomoxir, mice were first dosed with etomoxir and CB-839 to establish a dosing safety profile. Although CB-839 treated mice displayed no overt toxicity, CB-839 and etomoxir-treated mice in a pilot dose study (at doses of etomoxir from 60 mg kg$^{-1}$ to 40 mg kg$^{-1}$ including both NCr nude mice and genetically engineered mice on a mixed background) became acutely ill within hours post-dosing with severe lethargy and signs of hypoperfusion with symptoms meeting end point criteria requiring euthanasia. On pathologic examination, no definitive cause of death could be ascertained. Further attempts at CB-839 and etomoxir combination treatment beyond these pilot dosing trials were not pursued. All mouse experiments were conducted in compliance with ethical regulations approved by the DFCI Institutional Animal Care and Use Committee (IACUC) under protocol number 10-055.

**Western blotting analysis.** After SDS–PAGE, proteins were transferred to Hybond-N Nitrocellulose (Amersham Biosciences, 45-001-227). Membranes were blocked in Tris-buffered saline (TBS) containing 5% non-fat dry milk and 0.1% Tween 20 (TBS-T), before incubation with the primary antibody overnight at 4 °C. The membranes were then washed with TBS-T followed by exposure to the appropriate horseradish peroxidase-conjugated secondary antibody (1:10,000) for 1 h and visualized on Kodak X-ray film using the enhanced chemiluminescence detection system (Thermo Scientific, PI80196). The following antibodies were used: GLS (Abcam, ab93434, 1:1,000), β-actin (Sigma, A5441, 1:5,000), Cleaved Caspase-3 (Cell Signaling, 9661, 1:1,000), ALDH1L2 (LifeSpan Biosciences, LS-C178510, 1:100), CTH (Santa Cruz Biotechnology, sc-374249, 1:100) and VCL (Millipore, AB6039, 1:5,000).

**Metabolite and CB-839 measurements.** Steady state metabolomics experiments were performed as previously described[9,15]. In brief, PDAC cell lines were grown to ∼80% confluence in growth media with CB-839 or DMSO for the indicated time points (DMEM, 4 mM Gln, 25 mM Gluc, 10% dialysed FBS) on 6 cm dishes in biological triplicate. A complete media exchange with indicated drug was done 4 h before collection. Metabolomics and CB-839 experiments in Figs 2a,c and 3e, Supplementary Fig. 2i, and 3i were performed as described[15]. In brief, cell lines or mouse tissues were homogenized in methanol:water (80:20) containing 10 mmol l$^{-1}$ $^{13}$C$_5$, $^{15}$N-Glu as the internal standard and analysed for metabolite levels by liquid chromatography-tandem mass spectrometry (LC/MS-MS) using the SCIEX API4000 (Applied Biosystems). Mouse tissue homogenates were also analysed for CB-839 levels using a similar method except that 50 nmol l$^{-1}$ carbamazepine was used as the internal standard. Metabolomics experiments in Figs 1h and 4b,c,e,f, Supplementary Fig. 1k, 3h, and Supplementary Data 1 were performed using LC/MS analysis[48]. In brief, LC/MS analyses were conducted on a QExactive benchtop orbitrap mass spectrometer equipped with an Ion Max source and a HESI II probe, which was coupled to a Dionex UltiMate 3000 UPLC system (Thermo Fisher Scientific, San Jose, CA, USA). External mass calibration was performed using the standard calibration mixture every 7 days. Polar metabolites were extracted from cells or tissues using 1 ml of ice-cold 80% methanol with 10 ng ml$^{-1}$ valine-d$_8$ (Cambridge Isotope Laboratories DLM-488-PK) AND 10 ng ml$^{-1}$ of Phenylalanine-d$_5$ (Cambridge Isotope Laboratories DLM-1258-5) as internal standards. After a 10 min vortex and centrifugation for 10 min at 4 °C at 10,000 g, samples were dried using a speed vacuum. Dried samples were stored at −80 °C and then resuspended in 100 μl water; 1 μl of each sample was injected onto a ZIC-pHILIC 2.1 × 150 mm (5 μm particle size) column (EMD Millipore). The mass spectrometer was operated in full-scan, polarity switching mode. Relative quantitation of polar metabolites was performed with XCalibur QuanBrowser 2.2 (Thermo Fisher Scientific) using a 5 ppm mass tolerance and referencing an in-house library of chemical standards. The quantity of the metabolite fraction analysed was adjusted to the corresponding protein concentration and cell count calculated upon processing a parallel 6 cm dish. To trace Gln metabolism (Supplementary Fig. 3e,f, Supplementary Data 3), PDAC cell lines were grown as above with the indicated treatments and then transferred into Gln-free DMEM (with 25 mM glucose) containing 10% dialysed FBS and 4 mM [U-$^{13}$C$_5$]Gln (Cambridge Isotope Labs) at the indicated time points in the flux analyses.

**GLS assay activity.** GLS activity was measured in homogenates prepared from tumours using a coupled biochemical assay monitoring Glu production with the NADPH-dependent enzyme Glu dehydrogenase as described previously[15].

**Immunohistochemistry.** Tissues were fixed in 10% formalin overnight and embedded in paraffin. Hematoxylin and eosin and immunohistochemical analysis was performed as described[49]. In brief, slides were deparaffinized in xylene and rehydrated sequentially in ethanol. For antibodies requiring antigen retrieval, slides were incubated in a pressure cooker in sodium citrate buffer for 15 min. Slides were then quenched in hydrogen peroxide (0.3–3%) to block endogenous peroxidase activity and then washed in phosphate-buffered saline (PBS). Slides were blocked in normal serum for 1 h at room temperature. Primary antibody was applied either overnight at 4 °C or for 1 h at room temperature then developed using vectastain Elite ABC kit (Vector labs PK-6100) or Elite M.O.M kit (Vector Labs PK-2200) and DAB (Vector labs SK-4100). Slides were counterstained with hematoxylin

(Vector Labs H-3401) and then dehydrated sequentially in ethanol, cleared with xylenes, and mounted with Permount (Fisher). The antibodies and dilutions were as follows: SMA (Dako, M0851) 1:500, Ki67 (Ventana Medical Systems, 790-4286) 1:100. For differentiation status studies, Hematoxylin and eosin-stained slides were scored in a blinded manner. Differentiation was categorized throughout the entire tumour in eight equal fields as well, moderately, poorly or undifferentiated based on a combination of ductal architecture or lack thereof, level of pleomorphic nuclei and number of mitoses. For quantification of IHC staining of tumours, SMA was quantified using 10, 20× fields from each tumour. The percent area of SMA was calculated using ImageJ[50] to highlight the positively stained regions and averaged for each tumour. Likewise, Ki-67 percentage was calculated from 5, 40× fields in each tumour by counting the ratio of positively stained nuclei to negatively stained nuclei.

**Quantitative proteomics.** Quantitative mass spectrometry-based proteomics were performed as previously described[51]. TMT isobaric reagents were from Thermo Scientific (Rockford, IL, USA). Water and organic solvents were from J.T. Baker (Center Valley, PA, USA). Cells were homogenized by 20 passes through a 21 gauge (1.25 in. long) needle in lysis buffer (8 M Urea, 200 mM 4-(2-hydroxyethyl)-1-piperazineethanesulfonic acid (HEPES) pH 8.5, 1× Roche protease inhibitors, 1× Roche PhosphoStop phosphatase inhibitors) at a protein concentration of ∼5 mg ml$^{-1}$. The homogenate was sedimented by centrifugation at 20,000 × g for 5 min at 4 °C. Proteins were subjected to disulfide bond reduction with 5 mM dithiotreitol (37 °C, 25 min) and alkylation with 10 mM iodoacetamide (room temperature, 30 min in the dark). Excess iodoacetamide was quenched with 15 mM dithiotreitol (room temperature, 15 min in the dark).

Chloroform–methanol precipitation of proteins from cells was performed prior to protease digestion. In brief, four parts neat methanol was added to each sample and vortexed, one part chloroform was added to the sample and vortexed and three parts water was added to the sample and vortexed. The sample was centrifuged at 10,000 rpm for 2 min at room temperature and subsequently washed twice with 100% methanol. Samples were resuspended in 200 mM HEPES pH 8.5 for digestion. Protein concentrations were determined using the bicinchoninic acid assay.

For each of the samples, 100 μg of protein was digested at 37 °C for 3 h with LysC protease at a 1:100 protease-to-protein ratio. Trypsin was then added at a 1:100 protease-to-protein ratio and incubated overnight at 37 °C. Approximately 50 μg of peptides from each sample were labelled with TMT reagent (100 μg). For cell line analysis, three biological replicates were labelled per time point as indicated.

TMT reagents (0.8 mg) were dissolved in anhydrous ACN (40 μl) of which 10 μl was added to the peptides (resuspended in 70 μl of 100 mM HEPES, pH 8.5) along with 20 μl of ACN to achieve a final ACN concentration of 30% v/v. Following incubation at room temperature for 1 h, the reaction was quenched with hydroxylamine to a final concentration of 0.3% v/v. The TMT-labelled samples were combined at a 1:1:1:1:1:1:1:1:1:1 ratio. The sample was acidified, vacuum centrifuged to near dryness and subjected to C18 SPE (Sep-Pak, Waters). Samples were separated using basic pH reversed-phase HPLC for protein-level analyses and then pooled into 12 fractions.

Data were collected using an Orbitrap Fusion mass spectrometer (Thermo Fisher Scientific) coupled with a Proxeon EASY-nLC 1000 LC pump (Thermo Fisher Scientific). Peptides were separated on a 100 μm inner diameter microcapillary column packed with 0.5 cm of Magic C4 resin (5 μm, 100 Å, Michrom Bioresources) followed by 35 cm of Accucore C18 resin (2.6 μm, 100 Å, ThermoFisher Scientific).

Peptides were separated using a 3 h gradient of 6–30% ACN in 0.125% formic acid with a flow rate of 300 nl min$^{-1}$. Each analysis used an MS$^3$-based TMT method as described previously[27]. Mass spectra were processed using a Sequest-based in-house software pipeline as described previously[51]. Protein quantitation values were exported for further analysis. Each reporter ion channel was summed across all quantified proteins and normalized assuming equal protein loading of all nine samples.

**Bioinformatics analysis.** For visualisation of the proteomic data sets, volcano plots were created showing log$_2$ of relative abundance of proteins in the compared groups and corresponding $P$ values transformed into $-\log_{10}$ values. Top or bottom 5% regulated proteins were denoted in Fig. 5a,b to highlight greater variance in CB-839-72 h versus CB-839-24 h data sets. All proteins with at least two peptides identified were subjected to one-way analysis of variance (one-way ANOVA) with a Bonferroni-adjusted $P$ value to filter for proteins that were significantly changed. Calculations were performed on log10-transformed values. *Post hoc* tests (Bonferroni-adjusted $t$-test) were used to identify proteins with significant pairwise differences. Principal component analysis of proteins that showed significant differences in ANOVA was used to visualize the data set in a two-dimensional coordinate space. In addition, as the proteomic analysis could be considered a hypothesis-generating step for further validation experiments, false discovery rates (FDRs) were calculated both for ANOVA $P$ values using the Benjamini–Hochberg procedure with pairwise tests performed for significant ($P<0.05$, FDR $<0.05$) peptides. Gene set enrichment analyses[52] was performed for proteomic data. For this analysis proteins were included where at least two peptides were detected

and protein names were translated to gene names with Biomart[53]. This analysis searches for the locations of proteins associated with a given biological function or pathway within the ranked list of all proteins present in the data set and calculates an enrichment score depending on the set being globally up- or downregulated. GSEA has been successfully used for proteomic data[28] and its reliance on validated and curated biological pathways made it a good tool for identifying processes dysregulated by CB-839 treatment and ones associated with resistance. A false discovery rate is calculated to account for the number of comparisons made. The canonical pathways MSigDB category (c2.cp.v5.2.symbols.gmt) was used for GSEA[52]. Gene sets enrichment results for CB-839 72 h versus DMSO analysis (from Experiment 1) are represented in Supplementary Data 5. The DMSO versus CB-839 72 h versus CB-839-R data sets were analysed with one additional comparison for gene set analysis (DMSO versus CB-839 72 h, DMSO versus CB-839-R and CB-839 72 h versus CB-839-R) (lists are tabulated in Supplementary Data 10). Enrichment maps[29] were constructed for gene sets that differed significantly with a FDR < 0.01 and an overlap coefficient > 0.25 for comparison of CB-839 treated MPDAC-4 cells for 72 h and untreated cells (Fig. 5d) and CB-839-R cells versus CB-839 treated cells for 72 h (Fig. 8c). For comparison of CB-839-R cells versus untreated cells we used a cutoff value for FDR < 0.05 and overlap coefficient > 0.1 owing to lower variability of these data sets (Fig. 8b). For Connectivity Map 2.0 (Cmap 2.0) analysis proteins that were up- (fold change (FC) > 1.5) or downregulated (FC < 0.67) in two comparisons: CB-839 24 h versus DMSO and CB-839 72 h versus DMSO were evaluated. The Connectivity Map 2.0 (Cmap 2.0) analysis platform correlates the observed pathway dysregulation with known effects of experimentally tested drugs on multiple cell lines[40]. As this was an exploratory analysis, FDR values calculated as above were disregarded. Furthermore for the MPDAC-4 DMSO versus CB-839 24 h and CB-839 72 h, all three available data sets were analysed separately and drugs displayed in Fig. 8e,f were identified in all three data sets. As only one data set was available for the MPDAC-4 DMSO versus CB-839 72 h and CB-839-R, only proteins with adjusted FDR < 0.05, FC > 1.5, FC < 0.67 (according to protocol selection shown in Fig. 8d) were analysed using Cmap 2.0 (Supplementary Data 11). Metaboanalyst was used to analyze proteomic data in combination with metabolomics data where indicated(Supplementary Data 8)[34].

**Lentiviral-mediated shRNA targets.** shRNA vectors were obtained from the RNA Interference Screening Facility of Dana Farber Cancer Institute or Addgene, sequences as follows: shGLS-1, GCACAGACATGGTTGGTATAT (TRCN0000051135); shGLS-2, GCCCTGAAGCAGTTCGAAATA (TRCN0000051136). shGFP: 5′-GCAAGCTGACCCTGAAGTTCAT-3′ (Addgene plasmid #30323).

**Chemicals.** U13C labelled L-Gln (Cambridge Isotope Laboratories CLM-1822-H-0.1), L-Gln (Sigma, G3126), Glu (Sigma, G1626). CB-839 (Calithera Biosciences), BSO (Cayman Chemical, #14484), Methotrexate (Sigma, #A6770), Albendazole (Sigma, #A0325100), 17-AAG (Sigma, #A8476), Etomoxir (Cayman Chemical, #11969), BPTES (Sigma, #SML0601), MG-132 (Sigma, SML1135).

**Flow cytometry.** CellRox assay was performed at indicated time points after CB-839 treatment. Cells were incubated with 5 μM CellRox Reagent (Life Technologies, C10444) for 30 min. Excess CellRox was removed by washing the cells with PBS, and labelled cells were then trypsinized, rinsed, resuspended in PBS and analysed by flow cytometry.

**GSH assay.** Total GSH and GSH/GSSG ratios were measured after treatment with CB-839 according to the manufacturer instructions (Promega, V6611). Data was normalized to additional wells that received identical conditions using CellTiter Glo (Promega, G7570).

**Quantitative PCR.** Total RNA was extracted using RNeasy RNA isolation kit (Qiagen, 74104) according to the manufacturer's instructions. Reverse transcription was performed from 2 μg of total RNA using SuperScript Vilo IV cDNA synthesis kit (Thermo Fisher, 11756050), according to the manufacturer's instructions. Quantitative RT–PCR was performed with SYBR Green dye (BioRad, 1725271) using a CFX96 Real Time System (BioRad, 1855195). PCR reactions were performed in triplicate and the relative amount of cDNA was calculated by the comparative CT method using the 18S ribosomal RNA sequences as a control.

**Primer sequences.** Sequences for qPCR primers are as follows (all murine): E-Cadherin(F): CTCCAGTCATAGGGAGCTGTC, E-Cadherin(R): TCTTCTGA-GACCTGGGTACAC, N-Cadherin(F): AGGCTTCTGGTGAAATTGCAT, N-Cadherin(R): GTCCACCTTGAAATCTGCTGG, Snail(F): CACACGCT-GCCTTGTGTCT, Snail(R): GGTCAGCAAAAGCACGGTT, Twist(F): GGA-CAAGCTGAGCAAGATTC, Twist(R): CGGAGAAGGCGTAGCTGAG, Vimen-tin(F): TCCACACGCACCTACAGTCT, Vimentin(R): CCGGGACCGGGTC-ACATA, CHOP(F): AAGCCTGGTATGAGGATCTGC, CHOP(R): TTCCTGG-

GGATGAGATATAGGTG, BiP(F): ACTTGGGGACCACCTATTCCT, BiP(R): GTTGCCCTGATCGTTGGCTA, 18S(F): GTAACCCGTTGAACCCCATT, 18S(R): CCATCCAATCGGTAGTAGCG.

**Statistical analysis.** For all data aside from proteomics data, statistical analysis was done using GraphPad PRISM and Excel software. No statistical methods were used to predetermine sample size. Survival curve statistical analysis was performed using the log-rank (Mantel–Cox) test. When comparing continuous variables between two groups to each other, a Student's t-test (unpaired, two-tailed) was performed. For Supplementary Fig. 2a,b, the frequency of secondary tumour development and metastases development was compared using a two-tailed Fisher's exact test. A type 1 error probability < 0.05 was considered as the threshold for statistical significance, with adjustments made for multiple hypothesis testing where necessary, as described in the bioinformatics section above. Proteomic data statistical values presented in Supplementary Data 4, 6, 9. Remainder of statistical values presented in Supplementary Data 12.

**Data availability.** The quantitative mass spectrometry-based proteomics RAW data files have been deposited in the Peptide Atlas database under the accession code PASS00985. The processed proteomics data are available in the Supplementary Information. The authors declare that all the other data supporting the findings of this study are available within the article and its Supplementary Information files and from the corresponding author upon reasonable request.

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

## Acknowledgements

We thank Calithera Biosciences for use of CB-839, LC-MS/MS based measurement of CB-839 tumour and plasma levels and GLS inhibition assays. We thank Elizaveta Freinkman for assistance with LC-MS/MS analysis via the Whitehead Institute Metabolite Profiling Core Facility. This work was supported by National Cancer Institute Grants R01CA157490, R01CA188048 and P01CA117969; ACS Research Scholar Grant RSG-13-298-01-TBG; NIH grant R01GM095567; and the Lustgarten Foundation to A.C.K. J.D.M. is supported by a Burroughs Wellcome Fund Career Award for Medical Scientist, Joint Center for Radiation Therapy Grant, a KL2/Catalyst Medical Research Investigator Training award (TR001100), and the Claudia Adams Barr Program for Innovative Cancer Research. J.A.P. is supported by NIDDK Grant K01 DK098285. W.F. is supported by the First TEAM of the Foundation for Polish Science.

## Author contributions

J.D.M., D.E.B and A.C.K. conceived the study and designed experiments. D.E.B. and J.D.M. performed all experiments except for the TMT-based quantitative proteomics experiments performed by J.A.P. B.M. and W.F. performed bioinformatics analysis of proteomic data. M.Q.R assisted with cell culture, growth curves, and flow cytometry analysis. C.S. assisted with IHC and metabolomics analysis. X.W. provided technical support for cell line work and orthotopic mouse experiments. A.S.W.S provided technical support for cell line work. G.C.C. assisted with analysis of the genetically engineered mouse model clinical trial pathology. S.P.G. provided proteomics software and analysis support. J.W.H. provided proteomics support. J.D.M, D.E.B. and A.C.K. analysed the data and wrote the manuscript. All authors edited the manuscript.

## Additional information

**Competing interests:** A.C.K. is a founder and has financial interests in Vescor Therapeutics, LLC. A.C.K. is an inventor on patents pertaining to Kras regulated metabolic pathways, redox control pathways in pancreatic cancer, targeting GOT1 as a therapeutic approach and the autophagic control of iron metabolism. A.C.K is on the SAB of Cornerstone Pharmaceuticals. J.D.M. is an inventor on a patent pertaining to the autophagic control of iron metabolism. All other authors declare no competing financial interests.

