## [Peer Review File · Nature Communications]

Reviewers' comments:

Reviewer #1 (Remarks to the Author):

Compensatory Metabolic Networks in Pancreatic Cancers Upon Perturbation of Glutamine Metabolism

In this manuscript, Mancias et al evaluated the effect of a recently developed glutaminase inhibitor in pancreatic cancer cell lines and xenograft models as a potential therapeutic strategy for PDAC. In this study, they found that glutaminase inhibition exhibited initial cytostatic effects but subsequent metabolic adaptations in the cancer cells in redox and fatty acid metabolic pathways rendered the drug ineffective upon long-term treatment. The team proposes combinatorial approaches based on metabolomics and proteomic studies to circumvent these adaptive responses.

Major points

1. The authors noted a very interesting observation that secondary tumors were formed in the pancreas 5 days post treatment, which is extremely rapid for the pancreas. How big were these tumors when first detected? How do the scans on day5 compare to pre-enrollment scans?
2. The authors noted a significant increase in metastasis upon glutaminase inhibition. Does the drug induce any changes in EMT markers or in cell migration?
3. Does treatment with BPTES give similar results to CB-839?
4. Are the effects reported KRAS dependent? Do the authors see the same adaptive mechanisms in KRAS wildtype cell lines such as BXPC3?
5. The authors reported a synergy between CB839 and BSO, does this work through inhibition of GSH synthesis or through indirect induction of Nrf2 activity? What is the effect of CB839 on shKeap1 cells? Is the synergy with BSO observed also with other small molecules that are known to induce oxidative stress? Eg, have the authors looked at the effect of erastin in combination with CB839?
6. Combination therapeutic studies were performed in subcutaneous transplant models, the microenvironment (redox and metabolism) of which is very different from orthotopic pancreatic tumors. Is the synergy observed in the subQ model recapitulated in the autochthonous model used in the earlier part of the study (Figure 2)?

Minor points

7. What media conditions were used in the 3D experiments?
8. Figure 6E came before 6D in the manuscript, please revise the order

Reviewer #2 (Remarks to the Author):

This manuscript describes studies evaluating the importance of newly developed glutaminase inhibitors and their impact on PDAC tumor progression. The study reveals a very important finding that while short term inhibition of tumor growth is observed, sustained inhibition of GLS causes compensatory metabolic rewiring that could potentially be linked to advancing secondary tumor progression. The data reveal very significant observations and will be of immense value to the field.

The manuscript is very well written. Experimentally, the authors have conducted a very comprehensive study that includes both in vitro and in vivo analyses as well as utilizing proteomics and metabolomics in combination to interrogate metabolic networks. Appropriate controls have been used.

Major comment:

Discussion needs improvement. Since the study is extensive and evaluates complex metabolic rewiring, the discussion should be improved to direct the readership towards more solid conclusions (more depth). A tremendous amount of data has been acquired, but the authors have not provided sufficient connectivity between the observations. The conclusions currently discussed in the results section can be combined to produce a metabolic network model to provide hypothesis based on the observations. Basically, what do the authors think is happening in response to sustained GLSi. Currently, the discussion lacks depth. While pathway annotation has been shown, the conclusions are too general and don't commit to specific directions. Improving the discussion would help the manuscript to have a more significant impact on the future directions of this field.

Reviewer #3 (Remarks to the Author):

This manuscript reports the effects of CB-839, a potent glutaminase inhibitor, pancreatic cancer cell lines and xenografts. It is a very well-written paper that describes the lack of efficacy of CB-839 and further delineates the metabolic re-wiring that sustains resistance. The authors found that branched chain fatty acid oxidation (i.e., of branched chain amino acids) and anti-oxidant pathways were increased in cells and tumors treated with CB-839 as determined by metabolomics studies. The authors further tested with the combination of CB-839 and BSO (an 'oxidant') would synergize and indeed documented the effectiveness of this combination in vitro and in vivo. Since fatty acid oxidation seemed to be increased with CB-839, they also tested etomoxir (FAO inhibitor through targeting CPT), but did not find synergy and in fact, this combination was lethal in vivo. Overall, this is a well-executed study that has much to offer to the literature regarding re-wiring of metabolism as a consequence of targeted metabolic therapy.

Minor.

1. The authors should provide in supplemental data usable excel lists of significantly altered proteins from the proteomic studies.
2. I suggest that the authors provide a visual abstract/summary of the metabolic re-wiring for the less initiated readers.

Reviewer #4 (Remarks to the Author):

The manuscript "Compensatory Metabolic Networks in Pancreatic Cancers Upon Perturbation of Glutamine Metabolism" addresses the consequences of inhibiting the enzyme Glutaminase (GLS) in pancreatic ductal adenocarcinomas (PDAC) in vivo. In contrast to inhibiting GLS with the small molecule CB-839 in vitro, it failed to reduce solid tumor size and rather promoted appearance of multiple tumors in the specific pancreatic cancer mouse model chosen.

Given the general interest to target a cancer-specific metabolome for therapeutic intervention in hopes to stop growth or even destroy cancer cells, the study is timely and important, specifically because it demonstrates a (not unexpected) flexibility of cancer cells to adjust to metabolic changes for example in response to GLS inhibition. However, with any drug, that was tested for high efficacy in vitro but does not hold up to its promises in vivo, it leaves the reader with the very specific knowledge that CB-839 alone is not the route to go in order to treat PDAC. This is an information that is highly important to researchers that are focused on Glutaminase inhibition as an approach to treat cancer.

The manuscripts' main strength is its exploration of metabolic and proteomic data to determine

which compensatory mechanisms step in place upon GLS inhibition. The interesting section of the manuscript describes “the adaptive mechanisms pancreatic cancer cells use” upon inhibition of GLS.

To this end the authors perform a TMT-based, quantitative proteomics and find GO terms of amino acid transport upon 24h, and lipid catabolism, oxidation-reduction reactions, glutathione metabolism and fatty acid metabolic processes enriched following 72h of treatment with CB-839 within the 5% most regulated proteins with two peptide counts and a p-value of less than 0.05. Proteomic data is analyzed appropriately for example including Benjamini-Hochberg (BH) correction. However, it remains unclear where the BH correction was used to identify significantly regulated proteins. The broadness of the GO terms finally reported make it difficult to the reader to judge the true significance of the proteomic findings. For example, the rationale deduced from proteomic data (and metabolomic as well as literature data) to target GCLC/GCLM with BSO is not due to a clear change in abundance of GCLC/GCLM in the proteomic experiment, but rather based on a holistic interpretation of metabolic and proteomic data as well as stereotyped GO terms and canonical pathways combined. Specifically, GSH is downstream of Glu in the pathway provided in the manuscript (Figure 5). Thus, - with Glu levels still being 3/4 of control and no regulation of GCLC/GCLM -, treatment with BSO appears to be not the inhibition of a pathway that compensates for GLS inhibition but rather an additive combination of two inhibitors targeting the same pathway (e.g. GSH synthesis). Therefore, the statement that “integration of proteomic and metabolic profiling” is indicative of “compensatory pathways that may have therapeutic utility” remains to be shown because the alternative attempt to target fatty acid metabolism (a pathway that is potentially more likely to compensate GLS inhibition) with etomoxir in CB-839 resistant cells was ineffective.

A mechanistic explanation of what compensates Glu levels upon inhibition of GLS is missing or not further followed up in the manuscript. Significantly regulated proteins in the proteomic experiments provide a starting point to validate their regulation for example in tumor samples in vivo in order to tackle a mechanistic approach. A strong, experimentally supported hypothesis of why GLS inhibition remains unsuccessful in vivo remains open. The discussion of the manuscript highlights the struggle of the authors with the fact that no protein in the proteomic data was verified in its significance of being regulated: “protein expression is also readily adaptable to clinical scenarios, where if one could define metabolic biomarkers these could be assessed by immunohistochemistry on tumor biopsies”. The authors do not highlight any protein identified as regulated as a potential biomarker based on their proteomic dataset nor explicitly target any of the significantly regulated proteins found in the study.

Major points

1. The authors noted a very interesting observation that secondary tumors were formed in the pancreas 5 days post treatment, which is extremely rapid for the pancreas. How big were these tumors when first detected? How do the scans on day5 compare to pre-enrollment scans?

The tumors ranged in size from 3-7 mm in size when detected. Given the presence of additional PanIN lesions in the pancreas in this mouse model, our hypothesis is that these secondary tumors arose from pre-existing late-stage PanIN. In mice with secondary tumor development, the pre-enrollment scans were essentially normal in the area where a secondary tumor subsequently developed. We agree these are important points and we are actively investigating the mechanism of secondary tumor initiation and how this may be related to inhibition of glutamine metabolism.

2. The authors noted a significant increase in metastasis upon glutaminase inhibition. Does the drug induce any changes in EMT markers or in cell migration?

This is an interesting point. We assessed a panel of primary tumors from vehicle and CB-839 treated mice for EMT markers using RT-qPCR. As shown below and in Supplementary Fig. 2k, the data was highly variable, likely reflective of significant cellular heterogeneity within the tumors (in addition to tumor cells, stromal cells are of varying abundance in tissues). To assess this in a more controlled experimental system, we have treated PDAC cells with CB-839 for short and longer time points and assessed the expression of a panel of EMT markers. As shown below and in Supplementary Fig. 2l there was minimal increase in EMT markers at multiple time points. While a classical EMT response was not identified as an explanation for increased metastases, alterations in extracellular matrix, integrin, and cell adhesion pathways were identified in our cell culture proteomics (Fig. 5d, Fig. 7b, Supplementary Tables 2, 3, 7, 8). At this point it is unclear whether these proteomic changes or the minimal increase in EMT markers as measured by RT-qPCR have any role in the metastasis phenotype but is an area for future investigation.

(Supplementary Fig. 2k) Markers of epithelial-mesenchymal transition (EMT) assessed by RT-qPCR from tumors harvested from LSL-Kras^{G12D}; p53 L/+, Pdx1-Cre mice treated with vehicle (n=3) or CB-839 (n=3) on the clinical trial displayed in Fig. 2d. Expression levels are normalized to 18S ribosomal RNA and presented as mean ± s.d. of 3 tumors tested in triplicate.

(Supplementary Fig. 2l) Markers of EMT assessed by RT-qPCR in MPDAC-4 cells treated as indicated. Expression levels are normalized to 18S ribosomal RNA and presented as mean ± s.d. of 3 independent wells from a representative experiment (of 3 experiments). Significance determined by t-Test for (k,l), * P<0.05, ** P<0.01, *** P<0.001, ns: non-significant, P>0.05.

3. Does treatment with BPTES give similar results to CB-839?

We have now treated MPDAC-4 line with BPTES and see similar results with a short term cytostatic effect and at longer-term points, the emergence of regrowth. This is shown below and is now included in Supplementary Fig. 1b and 3c.

(Supplementary Fig. 1b) Cell proliferation dose-response curve for MPDAC-4 cell line treated with BPTES for 72 h. Error bars depict \pm s.d. of 3 independent wells from a representative experiment (of 3 experiments).

(Supplementary Fig. 3c) Relative proliferation of MPDAC-4 cell line treated long-term with BPTES or DMSO. Arrow represents time point when treatment was refreshed. Error bars depict \pm s.d. of 3 independent wells from a representative experiment (of 3 experiments).

4. Are the effects reported KRAS dependent? Do the authors see the same adaptive mechanisms in KRAS wild type cell lines such as BxPC3?

We have utilized the KRas wild-type BxPC-3 cell line as the reviewer suggested for growth assays with CB-839. As shown below (and in new Supplementary Fig. 1h and 3b), these cells behave similarly to KRas mutant cells that were previously included in the study. On the left is BxPC-3 cells treated short-term with CB-839 versus DMSO. On the right are BxPC-3 cells treated with CB-839 versus DMSO showing adaptive regrowth at later time points.

(Supplementary Fig. 1h) Relative proliferation of BxPC-3 cell line treated with CB-839 or DMSO. Data are plotted as mean relative cell proliferation \pm s.d. of 4 independent wells from a representative experiment (of 3 experiments). Significance determined by t-Test comparing last time point. *** $P < 0.001$.

(Supplementary Fig. 3b) Relative proliferation of BxPC-3 cell line treated long-term with CB-839 or DMSO. Arrow represents time point when treatment was refreshed. Error bars depict \pm s.d. of 3 independent wells from a representative experiment (of 3 experiments).

5. The authors reported a synergy between CB839 and BSO, does this work through inhibition of GSH synthesis or through indirect induction of Nrf2 activity? What is the effect of CB839 on shKeap1 cells? Is the synergy with BSO observed also with other small molecules that are known to induce oxidative stress? Eg, have the authors looked at the effect of erastin in combination with CB839?

This is an excellent point to extend our findings. As the reviewer suggests, we first assessed total glutathione levels as well as the ratio of reduced to oxidized glutathione (GSH:GSSG) upon CB-839 treatment. As shown below (and in updated Fig. 6b) there was a decrease in glutathione production and decrease in reducing capacity with acute CB-839 treatment but a return to baseline in CB-839-resistant cells.

(Fig. 6b) Total glutathione levels (left) and ratio of reduced to oxidized (right) after CB-839 treatment. Data are presented as mean \pm s.d. of 3 independent wells from a representative experiment (of 3 experiments). Significance determined by t-Test. * $P < 0.05$, ** $P < 0.01$, *** $P < 0.001$, ns: non-significant, $P > 0.05$.

To assess the Nrf2/Keap1 axis, we analyzed our proteomic data to assess the impact on Nrf2 targets. As below, this shows a variable response in Nrf2 targets with some upregulated at early time points and downregulated at later time points and the converse as well (Supplementary Fig. 4c). Together this suggests a global effect on redox homeostasis as shown with glutathione production decreases and that a Nrf2 response may be contributing to this as well.

(Supplementary Fig. 4c) Heatmap of Nrf2 pathway proteins involved in oxidative stress response, data plotted from Supplementary Table 2, MPDAC-4 Experiments 1-3. Values presented are the mean of $\text{Log}_2(\text{CB-839 treated sample} / \text{control})$ values from all three experiments.

To demonstrate synergy with oxidative stress in addition to what we have shown with BSO, we treated cells with CB-839 in combination with hydrogen peroxide. This confirms the synergy between CB-839 and redox stress that we saw previously with BSO and is now included as Fig. 6d (and below). While an excellent suggestion, we have not assessed Erastin in this system, but agree this is something worth investigating in future studies.

(Fig. 6d). Relative proliferation of MPDAC-4 cell line treated with CB-839 or DMSO with or without hydrogen peroxide (H_2O_2). Arrow represents time point when treatment was refreshed. Error bars depict \pm s.d. of 3 independent wells from a representative experiment (of 3 experiments). Significance determined by t-Test comparing last time point. ** $P < 0.01$.

- Combination therapeutic studies were performed in subcutaneous transplant models, the microenvironment (redox and metabolism) of which is very different from orthotopic pancreatic tumors. Is the synergy observed in the subQ model recapitulated in the autochthonous model used in the earlier part of the study (Figure 2)?

We agree with the reviewer regarding the benefits of using autochthonous models. While repeating the combination therapy studies with BSO that we showed to be effective in transplant models is an excellent idea, we hope the reviewer can appreciate that to generate a sufficient amount of genetically engineered mice with tumors would not be able to be accomplished in a suitable timeframe. To address the reviewers important point regarding the use of autochthonous models, we assessed whether we saw changes in markers of ROS-related damage (γH2AX and 4-HNE: 4-Hydroxynonenal) in vehicle versus CB-839 treated animals at the end of the clinical trial. Our expectation was that compensatory changes similar to those seen in our proteomic datasets would limit any change in markers of ROS-related damage. As shown below, there was no appreciable change in the level of γH2AX and 4-HNE staining in vehicle versus CB-839 treated autochthonous mice. This is consistent with our findings that these tumors develop compensatory changes that allow them to cope with redox stress. We agree that this is an important area of future research to understand what are the in vivo microenvironmental factors influencing redox and metabolism and how these are altered in the setting of glutaminase inhibition.

(Left) γ H2AX staining of tumors from LSL-Kras^{G12D}; p53 L/+, Pdx1-Cre mice treated with vehicle or CB-839 from clinical trial described in text and Fig. 2d. Tumors are from mice treated with CB-839 or vehicle for at least 2 weeks, scale bar 100 μ m. **(Right)** 4-HNE (4-Hydroxynonenal) staining of tumors as described on the left, scale is same as in γ H2AX figure.

Minor points

7. What media conditions were used in the 3D experiments?

Cells were grown on ultra low attachment plates in 2% matrigel supplemented with complete media (DMEM) as in Debnath et al. Methods. 2003; 30(3), 256-68. The methods have been updated with these media conditions.

8. Figure 6E came before 6D in the manuscript, please revise the order

Fig. 6 has been updated in multiple places, the order in the figure now reflects the order in the manuscript.

Reviewer #2:

This manuscript describes studies evaluating the importance of newly developed glutaminase inhibitors and their impact on PDAC tumor progression. The study reveals a very important finding that while short term inhibition of tumor growth is observed, sustained inhibition of GLS causes compensatory metabolic rewiring that could potentially be linked to advancing secondary tumor progression. The data reveal very significant observations and will be of immense value to the field.

The manuscript is very well written. Experimentally, the authors have conducted a very comprehensive study that includes both in vitro and in vivo analyses as well as utilizing proteomics and metabolomics in combination to interrogate metabolic networks. Appropriate controls have been used.

We thank the reviewer for the positive feedback

Major comment:

Discussion needs improvement. Since the study is extensive and evaluates complex metabolic rewiring, the discussion should be improved to direct the readership towards more solid conclusions (more depth). A tremendous amount of data has been acquired, but the authors have not provided sufficient connectivity between the observations. The conclusions currently discussed in the results section can be combined to produce a metabolic network model to provide hypothesis based on the observations. Basically, what do the authors think is happening in response to sustained GLSi. Currently, the discussion lacks depth. While pathway annotation has been shown, the conclusions are too general and don't commit to specific directions. Improving the discussion would help the manuscript to have a more significant impact on the future directions of this field.

We thank the reviewer for this great suggestion. To address these concerns, we have collaborated with the Fendler group and performed a more substantial and integrative analyses of our data (see revised description of proteomic results, discussion, revised Fig. 5, and new Fig. 7 as well as Supplemental Fig. 4 and 5, Supplementary Tables 2-5, 7-8). We include a portion of the expanded discussion section regarding the acute and sustained response to GLSi here:

We characterize the acute response to GLSi with CB-839 mediating cell survival and the sustained response that mediates resistance and allows for recovery of proliferation (see summary Supplementary Fig. 6). The acute response to GLSi is marked by induction of multiple stress response pathways including the ER stress response and anti-oxidant stress response. As a result of these and other responses, DNA synthesis, transcription, translation, and protein folding are attenuated precipitating the observed decrease in proliferation. Another major area of acute adaptation is in re-wiring cellular metabolism. Alterations in metabolic enzymes, including increased expression of pyruvate carboxylase (PC) can provide carbon to the TCA cycle via conversion of pyruvate to oxaloacetate and has been shown to be important for glutamine-independent cell growth^{40,41}. Likewise, proteins associated with glycolysis and oxidative phosphorylation are upregulated in response to CB-839 suggesting compensatory attempts by altering central carbon metabolism. Activation of lipid biosynthetic pathways, in part via PPAR γ signaling, supports findings that glutamine is an important source for accumulation of fatty acids and that alternative pathways are necessary in response to GLSi acutely³². Nucleotide biosynthesis reactions are affected by GLSi contributing to a decrease in DNA synthesis. Amino acid metabolism is likewise affected acutely by GLSi likely given a decrease in glutamate available for transamination reactions. CB-839 treated cells also appear to respond by upregulating a number of amino acid transporters to compensate as determined by proteomic and metabolomics measurements. Finally, multiple enzymes capable of providing glutamate via glutamine-dependent and glutamine-independent processes are upregulated including ASNS, BCAT1, GPT2, GGH, and OPLAH, all replenishing glutamate levels in MPDAC-4 cells. This final adaptation in the acute phase is likely responsible for MPDAC4 cells regaining proliferation. However, PATU-8988T cells never reestablish glutamate levels yet are able to continue proliferation so they likely utilize alternative pathways that may not be glutamine-dependent. CB-839-resistant cells maintain an elevated oxidative stress response, an increase in lysosomal processes, and upregulated glycolysis, nucleotide, sugar, amino acid, and pyruvate metabolism. These resistant cells also appear to operate at a new basal level of ER stress and as such upregulate protein folding capacity to compensate for proteotoxic stress in comparison to acutely treated cells. Multiple interesting avenues of investigation will stem from this comprehensive proteomic analysis. It will be informative to compare proteomic responses across additional pancreatic cancer cell lines as well as additional GLSi-sensitive and insensitive cancers to understand what are the conserved proteomic responses that may direct combination therapy.

32. Altman, B. J., Stine, Z. E. & Dang, C. V. From Krebs to clinic: glutamine metabolism to cancer therapy. *Nat. Rev. Cancer* 16, 619–634 (2016).
40. Cheng, T. *et al.* Pyruvate carboxylase is required for glutamine-independent growth of tumor cells. *Proc. Natl. Acad. Sci. U. S. A.* 108, 8674–9 (2011).
41. Sellers, K. *et al.* Pyruvate carboxylase is critical for non-small-cell lung cancer proliferation. *J. Clin. Invest.* 125, 687–698 (2015).

Of note, we have also now used the Connectivity Map 2.0 from the Broad Institute to determine if the observed proteomic changes can suggest additional efficacious therapeutic combinations. We have validated several of these in PDAC cells lines (below and Fig. 7i, j, k). All of this data and analysis is included in an expanded discussion section as well as a summary figure (below and Supplementary Fig. 6) that will provide a basis for future investigation in the field.

(Supplementary Fig. 6) Top: acute CB-839 inhibition leads to an initial significant decrease in glutamine derived metabolic pathways resulting in an increase in oxidative stress, proteotoxicity, and an integrated stress

response/endoplasmic reticulum (ER) stress response. This precipitates a significant cellular response including importantly a marked decrease in proliferation. Blue = downregulated metabolites or processes, Red = upregulated metabolites or processes, White = unchanged. This schematic represents a composite of data derived from this study including metabolomics measurements, proteomics, GSEA, and Connectivity Map 2.0 (CMAP) analysis. **Bottom:** prolonged exposure to CB-839 leads to resistance pathways that allow PDAC cells to reestablish proliferation. Notably, multiple glutamate-producing enzymes (BCAT1, OPLAH, ASNS, GGH, and GPT2) likely contribute to a reaccumulation of baseline levels of Glutamate (Glu) in MPDAC-4 cells. While glutathione and reactive oxygen levels are restored to baseline levels, CB-839 resistant cells remain sensitive to oxidant-inducing drugs (BSO, MTX). According to CMAP analysis, RT-qPCR, and 17-AAG inhibitor results, the ER stress response is less active in CB-839-R cells (indicated by a lighter shade of red). Likewise, given proteasomal inhibition is less effective in CB-839-R cells, this is also indicated with a lighter shade of red. Abbreviations: FAO: Fatty acid oxidation, TCA Cycle: Tricarboxylic acid cycle, GLS: Glutaminase, α -KG: α -ketoglutarate, UDP-GlcNAc: Uridine diphosphate N-acetylglucosamine, ER stress (Endoplasmic reticulum stress), ISR: Integrated stress response, BSO: L-Buthionine-(S,R)-sulfoximine, MTX: Methotrexate, ROS: Reactive oxygen species, GPT2: alanine aminotransferase 2, BCAT1, branched chain aminotransferase 1 OPLAH: 5-oxoprolinase ASNS: asparagine synthetase, GGH: Gamma-Glutamyl hydrolase.

Reviewer #3:

This manuscript reports the effects of CB-839, a potent glutaminase inhibitor, pancreatic cancer cell lines and xenografts. It is a very well-written paper that describes the lack of efficacy of CB-839 and further delineates the metabolic re-wiring that sustains resistance. The authors found that branched chain fatty acid oxidation (i.e., of branched chain amino acids) and anti-oxidant pathways were increased in cells and tumors treated with CB-839 as determined by metabolomics studies. The authors further tested with the combination of CB-839 and BSO (an 'oxidant') would synergize and indeed documented the effectiveness of this combination in vitro and in vivo. Since fatty acid oxidation seemed to be increased with CB-839, they also tested etomoxir (FAO inhibitor through targeting CPT), but did not find synergy and in fact, this combination was lethal in vivo. Overall, this is a well-executed study that has much to offer to the literature regarding re-wiring of metabolism as a consequence of targeted metabolic therapy.

We appreciate the reviewers enthusiastic comments.

Minor.

1. The authors should provide in supplemental data usable excel lists of significantly altered proteins from the proteomic studies.

These Excel tables have now been added as Supplementary Tables 2, 4 and 7.

2. I suggest that the authors provide a visual abstract/summary of the metabolic re-wiring for the less initiated readers.

We thank the reviewer for this excellent suggestion. This has now been included as Supplementary Fig. 6 and shown above in the response to reviewer #2.

Reviewer #4:

Given the general interest to target a cancer-specific metabolome for therapeutic intervention in hopes to stop growth or even destroy cancer cells, the study is timely and important, specifically because it demonstrates a (not unexpected) flexibility of cancer cells to adjust to metabolic changes for example in response to GLS inhibition. However, with any drug, that was tested for high efficacy in vitro but does not hold up to its promises in vivo, it leaves the reader with the very specific knowledge that CB-839 alone is not the route to go in order to treat PDAC. This is an information that is highly important to researchers that are focused on Glutaminase inhibition as an approach to treat cancer.

We thank the reviewer for their positive comments.

The manuscripts' main strength is its exploration of metabolic and proteomic data to determine which compensatory mechanisms step in place upon GLS inhibition. The interesting section of the manuscript describes “the adaptive mechanisms pancreatic cancer cells use” upon inhibition of GLS.

To this end the authors perform a TMT-based, quantitative proteomics and find GO terms of amino acid transport upon 24h, and lipid catabolism, oxidation-reduction reactions, glutathione metabolism and fatty acid metabolic processes enriched following 72h of treatment with CB-839 within the 5% most regulated proteins with two peptide counts and a p-value of less than 0.05. Proteomic data is analyzed appropriately for example including Benjamini-Hochberg (BH) correction. However, it remains unclear where the BH correction was used to identify significantly regulated proteins.

We have now collaborated with the Fendler group for a more in-depth bioinformatics and statistical analysis of the proteomic data as displayed in a revised Fig. 5 and new Fig. 7. We have now clarified the use of the Benjamini-Hochberg corrections in our Methods section and as follows. When examining the change of any one specific protein from our proteomics data, we use Benjamini-Hochberg correction values to establish significance of the observed changes. Our new analysis also includes Gene Set Enrichment Analysis (GSEA) that does not require Benjamini-Hochberg corrections prior to GSEA analysis. For exploratory Connectivity Map analysis when multiple datasets were available (such as Supplementary Table 2), we included proteins with at least two peptides detected and a fold-change criteria (>1.5 , <0.67) in the analysis and subsequently filtered based on overlap of drug pathways (see schematic in Fig. 7d and below). As only one dataset was available for the MPDACC-4 DMSO vs. CB-839-72H and CB-839-R, we only included proteins that met fold change thresholds (>1.5 , <0.67) and Benjamini-Hochberg calculated $FDR < 0.05$.

(Fig. 7d) Connectivity map (CMAP) workflow schematic for identification of candidate drugs for CB-839 synergy and pathway analysis.

The broadness of the GO terms finally reported make it difficult to the reader to judge the true significance of the proteomic findings. For example, the rationale deduced from proteomic data (and metabolomic as well as literature data) to target GCLC/GCLM with BSO is not due to a clear change in abundance of GCLC/GCLM in the proteomic experiment, but rather based on a holistic interpretation of metabolic and proteomic data as well as stereotyped GO terms and canonical pathways combined. Specifically, GSH is downstream of Glu in the pathway provided in the manuscript (Figure 5). Thus, - with Glu levels still being 3/4 of control and no regulation of GCLC/GCLM -, treatment with BSO appears to be not the inhibition of a pathway that compensates for GLS inhibition but rather an additive combination of two inhibitors targeting the same pathway (e.g. GSH synthesis). Therefore, the statement that “integration of proteomic and metabolic profiling” is indicative of “compensatory pathways that may have therapeutic utility” remains to be shown

because the alternative attempt to target fatty acid metabolism (a pathway that is potentially more likely to compensate GLS inhibition) with etomoxir in CB-839 resistant cells was ineffective.

To address these concerns, we have collaborated with the Fendler group as above to generate a more integrated and in-depth proteomic analysis. This has allowed us to make more definitive conclusions with respect to the proteomic findings and generate hypotheses for further testing (see above, Fig. 5 and 7). In addition to targeting the CTH pathway with BSO and fatty acid metabolism with etomoxir as part of a rational combinatorial approach, we also used methotrexate to target folate metabolism as it relates to redox homeostasis given a persistent elevation in ALDH1L2 (below and Fig. 6g, h, Supplementary Fig. 4i). Acute combinatorial treatment and treatment of CB-839-R cells (CB-839 resistant cells treated with CB-839 for greater than 15 days that resumed proliferative ability) with methotrexate was effective.

(Fig. 6g) Relative proliferation of MPDAC-4 cell line treated with CB-839 or DMSO with or without methotrexate (MTX). Data are plotted as mean relative cell proliferation, error bars depict \pm s.d. of 3 independent wells from a representative experiment (of 3 experiments). Significance determined by t-Test *** $P < 0.001$.

(Fig. 6h) Relative proliferation of CB-839 resistant MPDAC-4 cell line treated with CB-839 alone or in combination with methotrexate. Data are plotted as mean relative cell proliferation, error bars depict \pm s.d. of 3 independent wells from a representative experiment (of 3 experiments). Significance determined by t-Test ** $P < 0.01$.

(Supplementary Fig. 4i) Lysates from MPDAC-4 and PaTu-8988T cells treated with DMSO or CB-839 (1 μ m) were analyzed using antibodies to ALDH1L2, CTH, ACTB (β -actin loading control for MPDAC-4), and VCL (Vinculin loading control for PaTu-8988T).

In addition, connectivity Map analysis allowed us to identify potential therapeutic combinations not immediately apparent from our initial gene set enrichment analysis that have subsequently been tested and shown to be effective (above and Fig. 7d-h).

(Fig. 7e) CMAP analysis CB-839-24h vs. DMSO. Data represent

(Fig. 7f) CMAP analysis CB-839-72h vs. DMSO.

(Fig. 7g) CMAP analysis CB-839-72h

(Fig. 7h) CMAP analysis CB-839-72h vs. DMSO.

mean connectivity score \pm s.d. as determined from CMAP analysis of Experiments 1-3 independently (see methods, see Supplementary Table 9).

Data represent mean connectivity score \pm s.d. as determined from CMAP analysis of Experiments 1-3 independently (see methods, see Supplementary Table 9).

vs. DMSO. Data represent mean connectivity score \pm s.d. as determined from CMAP analysis of Experiment 4 (see methods, see Supplementary Table 9).

Data represent mean connectivity score \pm s.d. as determined from CMAP analysis of Experiment 4 (see methods, see Supplementary Table 9).

A mechanistic explanation of what compensates Glu levels upon inhibition of GLS is missing or not further followed up in the manuscript. Significantly regulated proteins in the proteomic experiments provide a starting point to validate their regulation for example in tumor samples in vivo in order to tackle a mechanistic approach. A strong, experimentally supported hypothesis of why GLS inhibition remains unsuccessful in vivo remains open.

We agree with the reviewer that this is an important point. There were a number of potential pathways that could compensate for reaccumulation of Glu levels in MPDAC-4 cells and may play a role in the in vivo setting. These have been illustrated in a Supplementary Fig. 4e, f and below. Specifically, we first examined changes in glutamine amidotransferases that could account for reaccumulation of fully labeled glutamate (M+5, Supplementary Fig. 3e). Among the glutamine amidotransferases, asparagine synthetase (ASNS) was significantly increased after CB-839 treatment suggesting ASNS may contribute to glutamate reaccumulation in the setting of GLSi (Supplementary Fig. 4e, Supplementary Table 5). Of note, ASNS has also been implicated in cell survival upon glutamine withdrawal via multiple mechanisms^{33,34}. While not a measure of flux, the remaining glutamine amidotransferases (CAD, PPAT, GMPS, CTPS1, CTPS2, GFPT1, GFPT2) were close to baseline measurements (Supplementary Table 5). Other glutamine-independent, glutamate-producing enzymes were elevated in MPDAC-4 cells including branched chain aminotransferase 1 (BCAT1), alanine aminotransferase 2 (GPT2), Gamma-Glutamyl hydrolase (GGH), and 5-oxoprolinase (OPLAH) suggesting alternate pathways for glutamate production not reliant on GLS (Supplementary Fig. 4f, Supplementary Table 5).

33. Zhang, J. *et al.* Asparagine plays a critical role in regulating cellular adaptation to glutamine depletion. *Mol. Cell* 56, 205–218 (2014).
34. Krall, A. S., Xu, S., Graeber, T. G., Braas, D. & Christofk, H. R. Asparagine promotes cancer cell proliferation through use as an amino acid exchange factor. *Nat. Commun.* 7, 11457 (2016).

(Supplementary Fig. 4e) ASNS is upregulated approximately 2- (Supplementary Fig. 4f) Glutamine-independent glutamate producing enzymes are upregulated in response to CB-839 in MPDAC-4 cells. Data

fold in response to CB-839 at 24 h. Data are derived from experimental data in Supplementary Table 2 and summarized in Supplementary Table 5.

are derived from experimental data in Supplementary Table 2 and summarized in Supplementary Table 5. For (e-f), mean fold-change \pm s.d. is presented, P values are calculated using a t-Test comparing normalized values of CB-839 treated samples vs. DMSO from Experiments 1-3, * P<0.05, ** P<0.01, *** P<0.001, ns: non-significant, P>0.05.

Importantly, we have provided a significantly expanded discussion and summary figure (Supplementary Fig. 6, above) of these pathways that we identified in the revised manuscript. The expanded data as well as the more extensive discussion will provide the field with several interesting opportunities to interrogate these pathways as part of future studies by our lab and others.

The discussion of the manuscript highlights the struggle of the authors with the fact that no protein in the proteomic data was verified in its significance of being regulated: “protein expression is also readily adaptable to clinical scenarios, where if one could define metabolic biomarkers these could be assessed by immunohistochemistry on tumor biopsies”. The authors do not highlight any protein identified as regulated as a potential biomarker based on their proteomic dataset nor explicitly target any of the significantly regulated proteins found in the study.

We thank the reviewer for pointing this out. To address the point, we validated the upregulation of CTH and ALDH1L2 in cell culture in response to CB-839 by immunoblotting (above and Supplementary Fig. 4i). Indeed, both of these were upregulated as was shown in the proteomic analysis. As potent inhibitors for CTH or ALDH1L2 were not available, we targeted the associated pathways to show the functional relevance of these upregulated proteins using clinically relevant inhibitors (Fig. 5f, Fig. 6e-h). We did attempt to target ALDH1L2 directly with RNAi-mediated knockdown; however, in the cells tested (PaTu-8988T) ALDH1L2 knockdown itself led to significant effects on viability and proliferation suggesting its importance basally (data not shown). While our data supports the conclusion that glutamine metabolism via GLS is important in vivo in pancreatic cancer models and that there exist adaptive responses to GLSi both in vitro and in vivo, we agree that future studies will be important to precisely define accurate predictive and prognostic biomarkers. These will allow us to define the best combination therapies and biomarkers for response. We have also tempered our language around this to highlight that this is a potential future benefit of such analyses but more validation is required.

REVIEWERS' COMMENTS:

Reviewer #1 (Remarks to the Author):

The authors have performed additional experiments to address my concerns, and include as clarifications and additions to the manuscript. The central premises are now better supported.

Reviewer #2 (Remarks to the Author):

The authors have provided a very detailed satisfactory effort to respond to the review comments. Specifically, the discussion is improved significantly, and the connectivity maps allow readers to better appreciate the data. No further improvement is necessary.

Reviewer #3 (Remarks to the Author):

The authors have extensively addressed key concerns from all reviewers.

Reviewer #4 (Remarks to the Author):

The comments of the reviewer were answered in completeness by the authors. The manuscript has gained overall in strength and length. Shortening the main body text might be beneficial.

Minor comment

In figure 5d and 7b,c: The content of the nodes is not disclosed in the figure and thus it is difficult to reconcile why these are displayed as individual nodes within each GO term "cloud". Collapse of nodes would be helpful or an annotation in a supplemental figure might be of advantage.

REVIEWERS' COMMENTS:

Reviewer #1 (Remarks to the Author):

The authors have performed additional experiments to address my concerns, and include as clarifications and additions to the manuscript. The central premises are now better supported.

We thank the reviewer for the positive comments.

Reviewer #2 (Remarks to the Author):

The authors have provided a very detailed satisfactory effort to respond to the review comments. Specifically, the discussion is improved significantly, and the connectivity maps allow readers to better appreciate the data. No further improvement is necessary.

We thank the reviewer for the positive comments.

Reviewer #3 (Remarks to the Author):

The authors have extensively addressed key concerns from all reviewers.

We thank the reviewer for the positive comments.

Reviewer #4 (Remarks to the Author):

The comments of the reviewer were answered in completeness by the authors. The manuscript has gained overall in strength and length. Shortening the main body text might be beneficial.

Minor comment

In figure 5d and 7b,c: The content of the nodes is not disclosed in the figure and thus it is difficult to reconcile why these are displayed as individual nodes within each GO term "cloud". Collapse of nodes would be helpful or an annotation in a supplemental figure might be of advantage.

We thank the reviewer for the positive comments. We have now shortened the main body text by ~800 words.

For Fig. 5d, 7b, and 7c we have now included the individual geneset analysis terms within each cloud within Supplementary Data 5 (for Fig. 5d) and Supplementary Data 10 (Fig. 7b, 7c).